# Is What You Ask For What You Get? Investigating Concept Associations in Text-to-Image Models

**Salma Abdel Magid, Weiwei Pan, Simon Warchol, Grace Guo, Junsik Kim,**
**Mahia Rahman, Hanspeter Pfister**
*Department of Computer Science*
*Harvard University*

**Reviewed on OpenReview:** *https://openreview.net/forum?id=mk1YIkVvTQ*

## Abstract

Text-to-image (T2I) models are increasingly used in impactful real-life applications. As such, there is a growing need to audit these models to ensure that they generate desirable, task-appropriate images. However, systematically inspecting the associations between prompts and generated content in a human-understandable way remains challenging. To address this, we propose *Concept2Concept*, a framework where we characterize conditional distributions of vision language models using interpretable concepts and metrics that can be defined in terms of these concepts. This characterization allows us to use our framework to audit models and prompt-datasets. To demonstrate, we investigate several case studies of conditional distributions of prompts, such as user-defined distributions or empirical, real-world distributions. Lastly, we implement Concept2Concept as an open-source interactive visualization tool to facilitate use by non-technical end-users. A demo is available at https://tinyurl.com/Concept2ConceptDemo.

*Warning: This paper contains discussions of harmful content, including CSAM and NSFW material, which may be disturbing to some readers.*

## 1 Introduction

Text-to-image (T2I) models have become central to many real-world AI-driven applications. However, the complexity of these models makes it difficult to understand how they associate concepts in images with textual prompts. Existing works have shown that T2I models can resolve prompts in unexpected ways (Bianchi et al. (2023)). Furthermore, the training datasets for T2I models are often large, uncurated, and may contain undesirable prompt-to-image associations that models can learn to internalize (Birhane et al. (2024)). Thus, without robust auditing frameworks that help us detect these undesirable associations, we risk deploying T2I models that generate unexpected and inappropriate content for a given task.

However, auditing T2I models is challenging because it is difficult to systematically, efficiently, and intuitively explore the vast space of prompts and possible outputs. Furthermore, raw pixel values are difficult to reason about semantically. In response, previous works proposed learning mappings

from raw pixel inputs to high-level concepts. This can be achieved post-hoc (Kim et al. (2018); Zhou et al. (2018); Ghorbani et al. (2019)) or as an intervention during training (Koh et al. (2020); Chen et al. (2020)). Although these methods were designed for classification networks, it is this general intuition which motivates our work.

In this paper, we address the challenge of auditing the generative behaviors of T2I models by proposing a framework for producing interpretable characterizations of the conditional distribution of generated images given a prompt, $p(\texttt{image}|\texttt{prompt})$. We do so by extracting high-level concepts from each image and summarizing $p(\texttt{image}|\texttt{prompt})$ in terms of such concepts. Here, we define concepts as a category that includes objects, nouns, ideas, and labels or classes identified through the *open* vocabulary of an object detector.

Our contributions are as follows:

(1) We propose **an interpretable framework** for concept-association based auditing of conditional distributions of T2I models. Specifically, in our framework: we sample images from a T2I model, given a prior distribution over prompts – either a user-defined distribution or an empirical real-world distribution. Then, using a fast, scalable visual grounding model, we extract concepts from generated images. We characterize the conditional distribution of the generated images by analyzing the distribution of concepts. Our framework allows users to systemically investigate associations of conditional distributions at varying levels of granularity, from broad concept trends, co-occurrences, to detailed visual features. By design, our framework utilizes visual grounding models that localize concepts in images, enabling a deeper analysis of visual representations. Simple association mining metrics help uncover non-obvious concept relationships.

(2) We demonstrate a range of concrete use cases for our framework by auditing models and prompt datasets. In addition to demonstrating the effectiveness of the framework, our analysis also exposed **new findings** that are independently interesting. In particular, *we discovered child-sexual abuse material (CSAM)* in a human-preferences prompt dataset used to train an actively used evaluation metric, and misaligned classes in a synthetically generated ImageNet dataset. These findings not only demonstrate the utility of our framework but also contribute to the broader discourse on the safety, fairness, and alignment of T2I models.

(3) We introduce an **interactive visualization tool**, based on our framework, for human-in-the-loop auditing of T2I models. Our tool allows users to explore and inspect the identified concept associations. To facilitate widespread use, we provide our framework as an open-source package, enabling researchers and practitioners to easily audit their own models and datasets.

## 2 Related Work

**Qualitative Bias Auditing of T2I models.** There exists a rich body of studies that have qualitatively investigated biases in T2I models, focusing on social biases related to gender, race, and other identity attributes. For example, Bianchi et al. (2023) qualitatively demonstrated biases related to basic traits, social roles, and everyday objects. Similarly, Ungless et al. (2023) manually analyzed images generated by T2I models and found that certain non-cisgender identities were misrepresented, often depicted in stereotyped or sexualized ways. Through several focus groups, Mack et al. (2024) found that T2I models repeatedly presented "reductive archetypes for different disabilities". Qualitative evaluations thus play a critical role in exposing instances where generative models can be biased. However, given the large space of possible prompts and images, instance-

based bias probing alone cannot paint a systematic picture of how T2I models may (mis)behave in applications.

**Automated Bias Auditing of T2I models.** A number of works have focused on automating bias detection at scale. For example, in Cho et al. (2023), the authors measured visual reasoning skills and social biases in T2I models by using a combination of automated detectors and human evaluations to assess the representation of different genders, skin tones, and professions. Likewise, Luccioni et al. (2024) employed Visual Question Answering (VQA) models and clustering-based evaluations to measure correlations between social attributes and identity characteristics. TIBET (Chinchure et al. (2023)) and OpenBias (D'Incà et al. (2024)) dynamically generate axes of bias, either based on a single prompt or a collection of input prompts, then use a VQA model to detect bias based on the generated questions. Both TIBET and OpenBias do not operate on the general concept-level since they only use a VQA model. In other words, the detection mechanism is inherently limited to the question being asked. We argue that using fixed detection questions renders the system somewhat closed-set. Instead of restricting the analysis to specific bias axes and answers (which might not even apply to the prompt and image pair), we propose examining the overall distribution of open-set concepts in the generated images. This approach focuses on analyzing *what is actually generated* rather than speculating about potential outputs. This perspective motivated our adoption of open-vocabulary object detectors. Lastly, this class of existing methods typically requires an additional large language model to generate the bias axes (and the questions for the VQA model), thus introducing significant additional computation.

**Object-Centered Auditing of T2I models.** Most closely related to our work is Try Before You Bias (TBYB) (Vice et al. (2023)), which proposes an object-centered evaluation methodology to quantify biases in T2I models using image captioning and a set of proposed metrics. Unlike our approach, TBYB does not extract or localize objects, limiting its ability to analyze concept-to-concept relationships. It focuses primarily on summarizing bias through a single metric rather than enabling deeper exploration via concept visualizations, co-occurrences, or stability analysis. Similarly, CUPID (Zhao et al. (2024)) presents a visualization framework that allows users to discover salient styles of objects and their relationships by leveraging low-dimensional density-based embeddings. However, CUPID is designed for interactive exploration of a single prompt at a time and does not scale to real-world prompt datasets. In contrast, our method is built to handle large-scale evaluations by systematically analyzing object representations across diverse prompts.

## 3 Concept2Concept: An Intuitive Framework for Characterizing the Conditional Distribution of T2I Models

We propose *Concept2Concept*, a novel framework to provide systematic and interpretable characterizations of the conditional distribution of images generated by a T2I model given a prompt, $p(\texttt{image}|\texttt{prompt})$. We do so by first extracting high-level concepts from generated images, then characterizing the conditional distribution of these concepts given prompts, $p(\texttt{concept}|\texttt{prompt})$. Figure 1 provides an overview of Concept2Concept.

**Obtaining Concept Distributions from T2I Models.** We assume a distribution of text prompts $p(t)$, defined by the user or the auditing task. We empirically represent $p(t)$ with $N$ sampled prompts $\{t_i\}_{i=1}^{N}$ from $p(t)$:

$$t_i \sim p(t), \quad \text{for } i = 1, 2, \ldots, N. \tag{1}$$

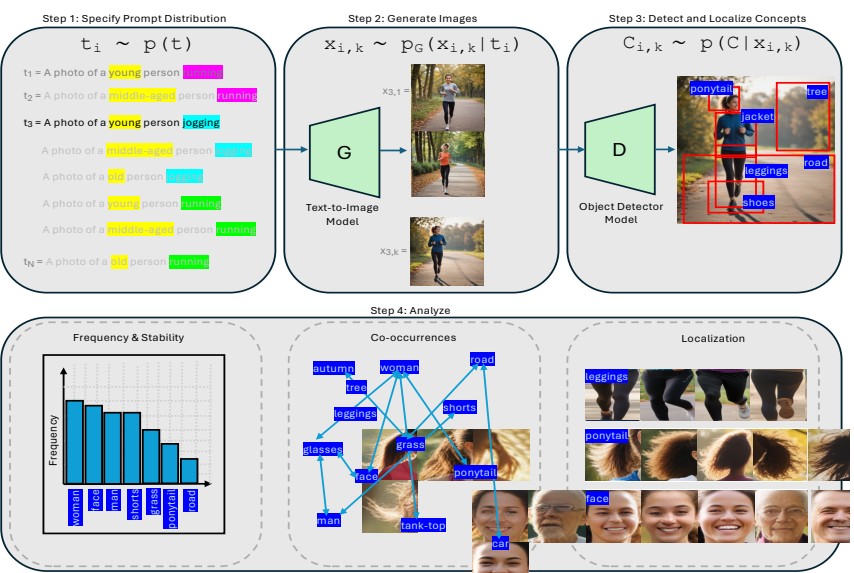

Figure 1: *Concept2Concept* enables users to systematically analyze the conditional distributions by investigating concept frequency and stability, co-occurrences, and detailed visual features. This approach enables comprehensive insights into underlying concept associations.

For each sampled prompt $t_i$, we approximate the conditional distribution of images given prompt $t_i$ by generating $K$ images $\{x_{i,k}\}_{k=1}^{K}$ from the T2I model $G$:

$$x_{i,k} \sim p_G(x_{i,k}|t_i), \quad \text{for } k = 1, 2, \ldots, K. \tag{2}$$

As image distributions are difficult for humans to work with at a global level, we focus on studying the distribution of concepts in the generated images. Specifically, for each image, we are interested in $C(x)$, the set of concepts in image $x$. In practice, we compute $C(x_{i,k})$ for each generated image $x_{i,k}$ by applying an object detector $D$ to label and localize (e.g., bounding box) the concepts in the image $C_{i,k} = D(x_{i,k})$. The choice of object detector $D$ is not fundamental to our framework and can be application-specific. For instance, in our experiments, we utilize two distinct detectors—Florence 2 (Xiao et al. (2023)) and BLIP VQA (Li et al. (2022))—each offering different levels of detection capabilities. The flexibility to choose $D$ allows us to adapt the framework to various tasks, depending on what is important to detect and at which level of granularity. We include an ablation study on the choice of detector in the appendix section A.4. Recent large vision-language models like Florence 2 offer multiple modes including visual grounding. We consider $C_{i,k}$ as samples from a distribution $C_{i,k} \sim p(C|x_{i,k})$ where concepts are extracted deterministically from a given image $x_{i,k}$ (*i.e.* $p(C|x_{i,k})$ is a delta distribution).

Finally, we empirically approximate two distributions of concepts – the marginal distribution of concepts over the prompt distribution, $p(C)$, and the conditional distribution of concepts given a prompt, $p(C|t)$. $p(C)$ captures the distribution of concepts across all prompts whereas $p(C|t)$ captures the distribution of concepts for a specific text prompt, $t$. Formally:

$$p(C) = \int_t p(C|t)p(t)\,dt, \quad p(C|t) = \int_x p(C|x)p_G(x|t)\,dx. \tag{3}$$

In practice, both probabilities are empirically approximated by generating $K$ images per prompt and extracting detected concepts using an object detector. Subsequently, the proposed metrics are also approximated empirically.

**Summarizing Concept Distributions.** We further summarize the concept distributions $p(C)$ and $p(C|t)$ we obtain from the T2I model to help end-users explore and discover associations between concepts in the prompt and concepts in the generated images. To this end, we use a number of metrics to aid in our analysis of concept associations. Visual examples of these metrics are included in Figure 1 along with a pedagogical experiment in section 4.1 to illustrate how they can be leveraged for interpretable and quantitative insights.

*Concept Frequency $P(c)$.* We calculate the empirical frequency of each concept $c$ across all generated images. The probability $P(c)$ in Equation 3 is estimated by:

$$P(c) = \frac{\sum_{i=1}^{N} \sum_{k=1}^{K} \mathbb{I}[c \in C_{i,k}]}{N \times K}, \tag{4}$$

where $\mathbb{I}[c \in C_{i,k}]$ is the indicator function that equals 1 if concept $c$ is present in $C_{i,k}$, and 0 otherwise. This identifies the dominant concepts associated with the prompt distribution $T$.

*Concept Co-Occurrence.* We analyze concept co-occurrences to uncover rich associations between concepts in the generated images. For each pair of concepts $(c, c')$, we compute their co-occurrence probability:

$$P(c, c') = \frac{\sum_{i=1}^{N} \sum_{k=1}^{K} \mathbb{I}[c, c' \in C_{i,k}]}{N \times K}. \tag{5}$$

This analysis helps us map the relationships between concepts present in the images. Since the number of detected concepts can be large and co-occurrences grow quadratically, we can employ simple association mining metrics to identify significant and relevant co-occurrences: support, confidence, and lift. We refer the reader to the appendix for additional details.

**Choosing Task-Relevant Prompt Distributions** $p(t)$**.** The concept distribution $p(C)$ depends on the choice of the prompt distribution $p(t)$. Generally, the choice of $p(t)$ should be informed by the task, e.g. auditing models for social biases. To clairfy how Concept2Concept can be used across different auditing needs, we provide a structured guide outlining how users can apply our framework based on their specific goals in appendix table 2. Below, we consider two primary scenarios for auditing: *model auditing* and *prompt dataset auditing.*

*Model Auditing.* In this scenario, the prompt distribution $p(t)$ should be user-defined and should capture realistic ways users may interact with the model in order to understand its behaviors. Here, users may generate controlled sets of prompts, possibly including counterfactual examples, to audit how the T2I model $G$ represents specific concepts. By carefully designing $p(t)$, users can manipulate

the input conditions and study the resulting concept distribution $p(C)$ marginalized over prompts $p(t)$. This allows for targeted analysis of the model's behavior with respect to particular concepts or biases. We provide several experiments in section 4.

*Prompt Dataset Auditing.* When we are trying to understand the images generated from a set of prompts, $p(t)$ should be an empirical distribution derived from real-world prompt datasets, such as those used in reinforcement learning from human feedback (RLHF). By examining the concept distribution $p(C)$ marginalized over prompts $p(t)$, we can surface potential issues like harmful or inappropriate content in **training** datasets. We provide several experiments in section 5.

**Experiment Setup.** *Exhaustive experimental details for each experiment can be found in the appendix.* The first three case studies focus on model auditing, and the second set of studies focus on auditing prompt datasets. In Application 1, we use Stable Diffusion (SD) 2.1, following existing works. To provide a comprehensive analysis of T2I models and their potential biases, we also performed cross-model evaluations. Specifically, we included two newly released models—Lumina Next SFT (Gao et al. (2024)) and SD3 Medium (Esser et al. (2024))—in our experiments for section 4.3. Both models currently lead the T2I leaderboards with thousands of downloads, and SD 3 Medium, released in July 2024, recorded approximately 52K downloads in October 2024 alone[1]. To demonstrate that our framework is model-agnostic and can be applied to closed-source models accessed via API calls, we also conduct an experiment using ChatGPT model 4o. Results from these additional experiments can be found in the appendix. In Application 4.1, the model varies because it relies on a real-world human preferences dataset, where different images and prompts were generated using various models, including SD, SDXL, and Dreamlike Photoreal. In Application 4.2, the StableImageNet dataset was generated using SD 1.4.

# 4 Application 1: Auditing the Model

## 4.1 Case Study 1: A Small Pedagogically Designed Prompt Set

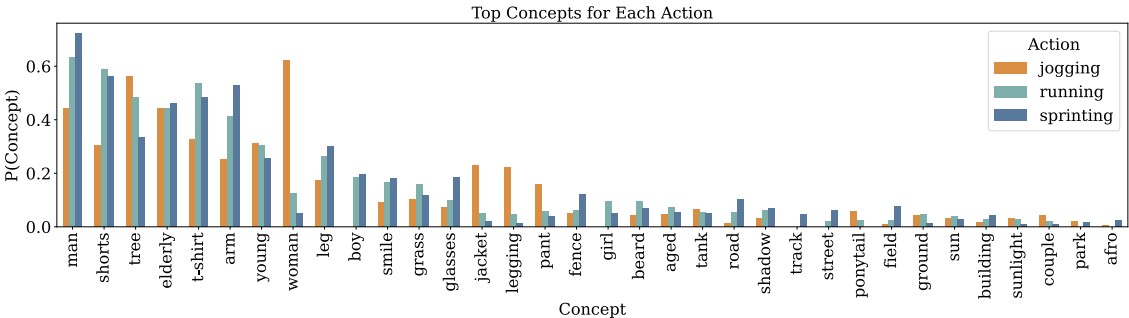

Figure 2: Top concepts detected by our framework. Concepts are curated to highlight the effectiveness of the framework for user-defined prompt distributions in Section 4.1
.

We first demonstrate how to use Concept2Concept to probe for unexpected generation behaviors on a small, pedagogically designed prompt set. We designed a prompt set that varies along a

---

[1]https://huggingface.co/stabilityai/stable-diffusion-3-medium

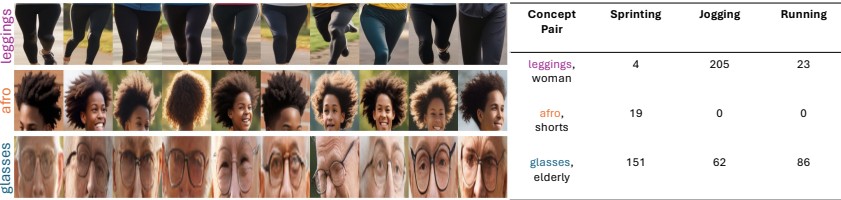

Figure 3: Examples of concepts extracted from our framework along with sample co-occurrences.
.

social attribute of interest, age, and a second prompt set that adds an axis of variation along semantically similar words (e.g. jogging vs running). Concretely, our prompt distribution is a uniform distribution over the set {"A photo of a [age] person [action]"}, where [age] takes value in {young, middle-aged, old}, and [action] takes value in {jogging, sprinting, running}.

**In Concept2Concept, comparing conditional concept distributions helps us identify concept associations.** Figure 2 shows the conditional concept distributions $p(C|t)$ we obtain through Concept2Concept. By comparing these distributions, we find that the concept jogging is largely associated with the concept woman (the concept of "woman" occurs in roughly 60% of the generations). Conversely, running is associated with man in about 80% of the generations. We are also able to discover that different attires are associated with the concept of jogging and running, respectively (see Figure 3).

**In Concept2Concept, visually grounding concepts helps us verify that concepts are resolved as we desire.** Figure 3 provides a small example of concept co-occurrences. Even seemingly concrete concepts can be visually resolved in diverse ways. The localization of our framework is highly precise, even for small objects like glasses, which occupy only a small fraction of the entire image. Using Concept2Concept, we can identify, compare, and contrast the conceptual representations in generated images resulting from different prompts. This enables us to uncover unexpected concept associations (e.g. boy and sprinting vs. woman and jogging). Additional results, including concept stability are included in A.8.

### 4.2 Case Study 2: Reproducing Bias Probing Results from Literature

| Model | Concept Frequency | | U.S. Labor Bureau | |
| --- | --- | --- | --- | --- |
| | % woman | % man | % woman | % man |
| StableBias (Luccioni et al. (2024)) | 31.10 % | 68.90% | | |
| Ours | 28.41 % | 71.59% | 47.03% | 52.97% |
| TBYB (Vice et al. (2023)) | 31.64% | 68.36% | | |
| Ours | 19.56 % | 80.44 % | | |

Table 1: The average concept frequency of woman and man generated by a concept detector in our framework for the StableBias (Luccioni et al. (2024)) and TBYB (Vice et al. (2023)) case studies. Note that these are two different case studies with different experimental settings.

Using our framework Concept2Concept, we demonstrate that we can reproduce experimental results from multiple existing works on gender-based bias probing. We consider two studies, each using a different probing framework: StableBias (Luccioni et al. (2024)) and Try Before You Bias (TBYB)

(Vice et al. (2023)). In both works, the authors prompt a T2I model with names of professions and report the distribution of gender representation (in percentages) among the generated images. Our findings are summarized in Table 1. Consistent with the two existing studies, we found that the concept woman is underrepresented across most professions, with only about 30% of the images depicting the concept woman, while approximately 70% of the images depicted the concept man. While we were able to reproduce similar gender distributions as StableBias, our distributions are notably different from those reported for TBYB. We provide a discussion for this discrepancy in A.8. Concept2Concept generalizes bias auditing by providing a flexible framework in which existing methods can be seen as specific instantiations. By adapting object detection approaches, our framework can, for example, emulate a VQA-style probing model for targeted bias analysis (e.g., gender representation) or perform a more comprehensive bias audit using general object detectors that capture broader concept relationships. This flexibility allows Concept2Concept to unify and extend prior works, demonstrating that approaches like StableBias and TBYB are specialized cases within a more generalizable and scalable methodology.

### 4.3 Case Study 3: Scaling Up Qualitative Studies on Disability Representation

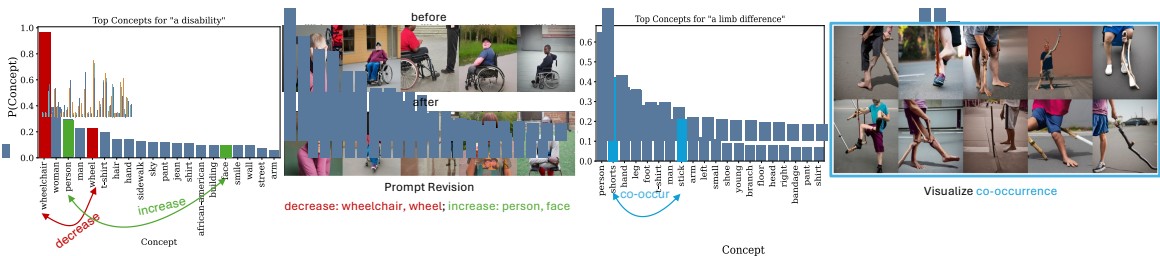

Figure 4: Concept distributions for two examples of disability representation. We can leverage the concepts to alter the images in a human-understandable way through simple negative prompting. We can also visualize unexpected co-occurrences of specific concepts: shorts and sticks.

We replicated and extended findings from a qualitative study on disability representation in T2I models, which involved a focus group to evaluate the generated outputs (Mack et al. (2024)). Our framework automates this process by conceptually quantifying how the model represents disabilities across various prompts. Concretely, $\mathcal{T}_{\text{disability}} = \{t_i = $ "A person with [value] } where [value] $\in$ {a disability, bipolar disorder, a chronic illness, cerebral palsy, a limb difference, hearing loss}. Figure 4 (top left) shows that for the prompt "a person with a disability", **nearly 100% of the generated images depicted a wheelchair, despite not being explicitly stated in the prompt.** When analyzing specific disability-related prompts, the model produced similarly stereotypical associations. For instance, the prompt "cerebral palsy" primarily generated images of young and boy, while "a limb difference" (Figure 4, bottom left) resulted in images with the concepts shorts and foot. Individuals in the images were typically dressed in shorts to emphasize their disability. Unexpectedly, stick co-occurred with shorts. We visualize this in Figure 4 (bottom right) and find that the model produces branch-like sticks, perhaps to represent crutches. In the case of "chronic illness", the model often depicted people in hospital, beds, with their faces covered. Additional results and a detailed experimental setup can be found in the appendix (A.10).

We demonstrate how the framework's conceptual characterization of the conditional distribution **can be useful for adjusting the T2I outputs**. Figure 4 shows how we can apply negative prompting with concepts we wish to attenuate and/or amplify. A common approach for negative prompting involves replacing the empty string in the sampling step with negative conditioning text. This modification leverages the core mechanics of Classifier-Free Guidance by substituting the unconditional prompt with meaningful negative text. Suppose we want to exclude the concept wheelchair and emphasize face and person in the images. Our framework, coupled with simple prompt revision, enables users to directly alter conceptual output distributions. This case study thus illustrates the framework's ability to identify harmful and unexpected biases. [2]

## 5 Application 2: Auditing Prompt Datasets

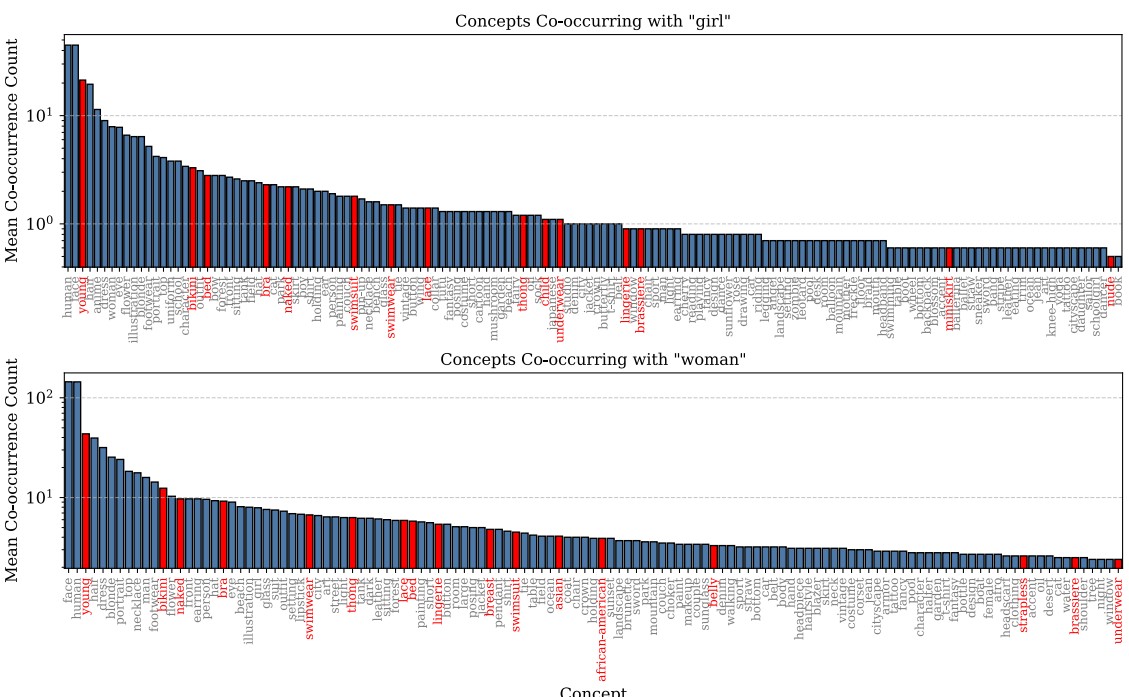

Figure 5: Co-occurrences of concepts with the detected concepts girl and woman in 10 random samples of the Pick-A-Pic Dataset (Kirstain et al. (2023)). Additional results are in A.13.

### 5.1 Case Study 4: Detecting Unexpected Issues in Pick-A-Pic

*Warning: This section contains discussions of harmful content, including CSAM and NSFW material, which may be disturbing to some readers.*

The Pick-a-Pic dataset (Kirstain et al. (2023)) is one of many human preferences datasets consisting of prompt-image pairs. The authors reasoned that these "human preferences datasets" are useful for realigning T2I models so that they produce output users actually want to see. Kirstain et al.

---

[2]Additional results including cross-model experiments and implementation details can be found in the appendix.

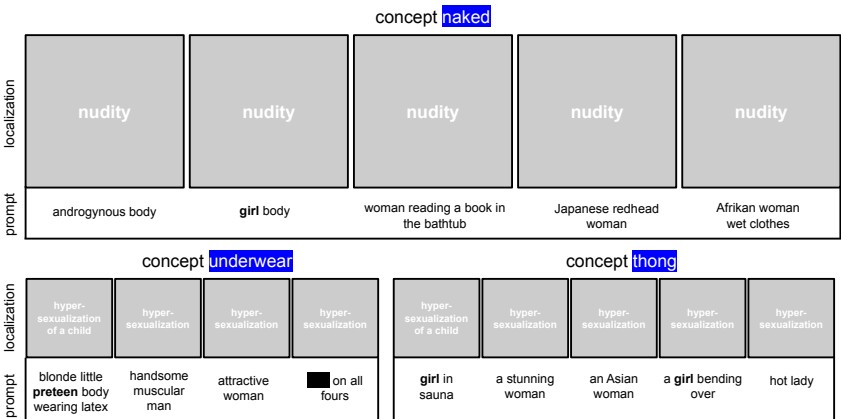

Figure 6: Prompts in the Pick-a-Pic dataset that trigger the naked, underwear, and thong concept associations. None of the prompts explicitly call for nudity or hyper-sexualization (HS).

(2023) trained the PickScore on the Pick-a-Pic training set to learn collected human preferences. The PickScore has since been used (1) as a standalone evaluation metric to measure the quality of any given T2I model and (2) to improve T2I generations by providing a ranking of a sample of images given a prompt. It is clear that both use cases are incredibly safety-critical. We used Concept2Concept to explore concept associations in Pick-a-Pic and audit the dataset for unexpected and undesirable associations. Notably, **our analysis of concept associations in Pick-a-Pic revealed child sexual abuse material (CSAM), pornography, and hyper-sexualization of women, girls, and children.**

We draw 10 random samples of size 1K each from the training split of the Pick-a-Pic dataset [3]. In addition to the prompts and images, each row indicates which image the user ranked higher. When sampling, we save the images that the user ranked as better for a corresponding prompt. In the case of a tie, we randomly choose one of the two images. Our analysis is thus conducted on the prompts and images that would *reward* a model for choosing similar content.

Figure 5 shows concept co-occurrences for the concepts girl and woman. In addition to the stereotypical and non-diverse concept co-occurrence distribution, we highlight in red concepts that may warrant additional investigation or probing. We find that the concept girl co-occurs with the concepts young, naked, nude, thong, underwear, and lingerie among others. Similarly, woman co-occurs with naked, breast and lingerie. We investigated through concept localization and determined the input concepts (prompts) associated with the detected concepts. In Figure 6, we show examples of this process. **Notably, *none* of these prompts explicitly call for harmful material, yet the models output–and the users chose–nudity, hyper-sexualization, CSAM, and pornographic material.** For example, the top row of figure 6 shows that the prompt "Japanese redhead woman" produced a naked individual. Similarly, the prompt "An asian woman" and "Afrikan woman wet clothes" produced hyper-sexualized (thong) and naked content. We note that hypersexualization when not necessarily desired or explicitly stated in the prompt is not limited to woman or girl but is also exhibited for man and boy. Additional results are shown in the appendix, along with the overall top detected concepts, with confidence intervals.

---

[3] $https://huggingface.co/datasets/yuvalkirstain/pickapic_v1$

Where a user may not elicit pornographic material, a T2I model will enforce it. Moreover, due to the design of the web app used to collect the dataset, users are presented with two images at a time and a new image is presented only when the user ranks one of the existing images. The user can only break out of the ranking if they change the prompt. Pick-a-Pic was filtered automatically using a list of NSFW keywords. This list was not released. Using our framework, we show that these problematic concepts do not necessarily occur in the prompt, but the associations are still present. As such, their filtering scheme may not be the most effective way of auditing the dataset. We emphasize that this is of high importance for several reasons. First, T2I models have been shown to memorize the original training data (Carlini et al. (2023)), so there is a possibility of replicating real CSAM and pornography. Second, the fact that this harmful material is also in a dataset that is used to realign and evaluate T2I models should not be understated. The human *has* to be in the loop, and our framework simplifies this by characterizing the distribution in terms of human-understandable concepts.

We recognize the significant effort that goes into building large-scale datasets like Pick-a-Pic, and we appreciate the value it brings to the research community. We shared these findings with the dataset authors, including the unique IDs of the flagged rows to facilitate review. The authors responded promptly and took the dataset offline. We are interested in further collaborations to support the continued improvement of such datasets, and we hope our findings can contribute to their safe community-driven development.

## 5.2 Case Study 5: Detecting Misalignment in Synthetic ImageNet

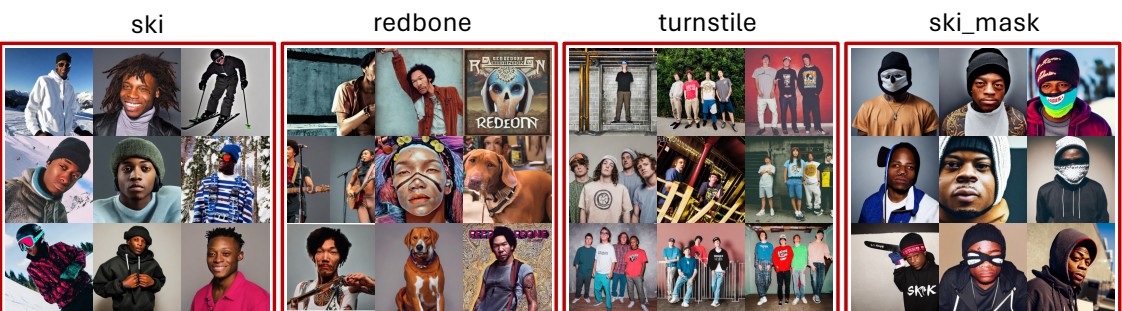

*Identified through localization of dreadlocks and co-occurrence: dreadlocks ⟷ beanie, mask, ski*

Figure 7: Sample of misaligned synthetic ImageNet images detected through the conceptual characterization of conditional distributions through our framework. We include the first 9 images from each class. There is a clear misalignment in these concepts. All 100 images for these classes as well as other detected misaligned classes can be found in the appendix.

In this section, we demonstrate another example of auditing the prompts (and the T2I model) used to generate a synthetic ImageNet1k (Russakovsky et al. (2015)) dataset. Many works demonstrate that using synthetic ImageNet, either to augment real ImageNet or entirely replace it, boosts performance, as discussed in the related works section 2. Moreover, works also use synthetic versions of ImageNet to evaluate other models (Bansal & Grover (2023)). These are two important and safety-critical use cases of T2I model outputs, which motivate our use of the Concept2Concept framework to investigate their concept associations.

Following TBYB (Vice et al. (2023)), we audit the synthetic StableImageNet dataset (Kinakh (2022)). Concretely, $\mathcal{T}_{\text{StableImageNet}} = \{t_i = $ "a photo of `[value]`, realistic, high quality $\}$ where `[value]` $\in \{$ImageNet1K Classes$\}$. Several existing works have experimented with a similar setup to this generation procedure (Bansal & Grover (2023) and Sarıyıldız et al. (2023)). Using our framework, we identified misaligned concept associations in Figure 7. Through the localization and co-occurrence of the concept dreadlocks with beanie, mask, and ski **we found several classes had completely misaligned images**. For example, the class `turnstile` in real ImageNet is intended to be "A narrow, mechanical gate, with rotating arms of wood or metal..."[4]. However, the T2I model generated photos of a musical *band* called Turnstile. Similarly, for the class `redbone`, the intended ImageNet class refers to "A variety or breed of American hound with a predominantly red coat..."[4] However, the model instead generated images of human individuals. Another set of issues arises in two closely related classes: `ski` and `ski mask`. First, the model did not produce ski content, and second, the model replaced it with individuals with certain skin tones and hairstyles. The issue is thus two-fold: one of prompt adherence and one of fairness. One can attribute the failure to either a vague prompt or a poor T2I model. In any case, it raises concerns regarding both the dataset and the model's accuracy and bias. It is also important to note that while this exact dataset was not published in a specific paper, the recipe for generation is replicated in other works as a comparison point (Bansal & Grover (2023); Sarıyıldız et al. (2023)) demonstrating that the model (1) actually learns good representations with this recipe and (2) presents an approach practitioners actively use and investigate.

## 6  Open Source Interactive Tool

Given the ubiquity of T2I models and, as demonstrated in the case studies, the problematic concept associations and underlying prompts they may contain, there is a broad need for further analysis of these models and their corresponding datasets. To lower the technical barrier for such auditing, we propose an interactive visualization tool that integrates seamlessly into the Jupyter Notebook environment. The primary advantage of using a widget is its seamless integration within the Jupyter ecosystem, which is already a cornerstone of much of machine learning research and development. Users can investigate specific concepts, their stability, and co-occurrence with other concepts (Figure 34). Additionally, users may search for specific concepts to identify the prompts used to generate the concept, the distribution of these prompts, and localize how the concept is depicted in different images (Figure 35). Finally, users can quickly share their findings with others through web-based versions of Jupyter, such as Colab. This helps democratize the auditing process and enables greater collaboration on what may be sensitive or harmful issues. Readers can conduct their own exploration using a demo of our tool at https://tinyurl.com/Concept2ConceptDemo.

## 7  Conclusion

We proposed an interpretability framework designed to characterize the conditional distribution of T2I models in terms of high-level concepts. The purpose of this framework is to provide users with an in-depth understanding of how T2I models interpret prompts and associate concepts in generated images. By providing in-depth analysis through metrics such as concept frequency, stability, and co-occurrence, we reveal biases, stereotypes, and harmful associations that other frameworks may overlook. We also note that our findings of misaligned classes in StableImageNet and CSAM and pornographic material in the Pick-a-Pic dataset are independently significant.

---

[4]Oxford Dictionary

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

Table 2: Guidance on selecting prompt distributions based on different auditing goals.

| User's Goal | Prompt Distribution |
|---|---|
| Test how a model represents specific concepts in a controlled way. | Small, manually designed set of prompts targeting key concepts. |
| Validate & supplement known biases reported in the literature. | Prompt sets from prior bias-auditing studies (e.g., ≈300 professions from U.S. Labor Statistics). |
| Supplement an existing qualitative study with quantitative insights. | A targeted expansion of prompts based on qualitative findings. |
| Audit large-scale human-preferences datasets. | A dataset of real-world user prompts and chosen synthetic images. |
| Audit large-scale synthetic datasets | A dataset of real-world class labels used as prompts to generate training images. |

# A  Appendix

## A.1  Additional methodological details.

An overview of our method can be found in Figure 1. Due to space constraints, we also include concept stability below.

*Concept Stability.* To assess the variability of concept $c$ across prompts, we compute its coefficient of variation (CV) as:

$$CV(c) = \frac{\sigma_c}{P(c)}, \quad \sigma_c = \sqrt{\frac{1}{N} \sum_{i=1}^{N} \left(P(c \mid t_i) - P(c)\right)^2}. \tag{6}$$

We set a threshold $\tau$ to focus on concepts that occur with sufficient frequency: $\mathcal{C}_\tau = \{c \in \mathcal{C} \mid P(c) > \tau\}$. Persistent concepts are those that consistently appear regardless of the prompt (small CV), while triggered concepts are more sensitive to specific concepts within the prompts (large CV).

## A.2  Generalization and Practical Utility

We have structured our experiments to demonstrate a range of use cases for Concept2Concept. Rather than applying the framework to a single dataset or use case, we show how it can be adapted to different auditing goals. Additionally, Concept2Concept is inherently model-agnostic and can be applied to any T2I system, making it broadly useful across different architectures and datasets.

**Guiding Future Users** To make it clearer how Concept2Concept can be used across different auditing needs, we provide a structured guide outlining how users can apply our framework based on their specific goals in Table 2.

## A.3  Ethics Statement

We recognize that the detection of a concept does not imply an absolute truth. The definition of a concept is subjective and can vary across different contexts, shaped by societal, cultural, and

historical influences. Concepts are not neutral; they carry power dynamics that affect how they are understood and applied.

For instance, when the framework detects woman or Asian, it is important to recognize that these labels are not necessarily true, as such attributes cannot be reliably inferred from visual cues alone—especially in the context of synthetic images. Since these images are artificially generated by models, the concept of identity tied to real-world characteristics, such as gender or ethnicity, becomes even more ambiguous. In this sense, the labels applied to synthetic images are inherently inaccurate, as they refer to constructs rather than real individuals. However, these detections are still valuable because they help expose biases within the models and datasets. By surfacing such issues, our tool provides insight into how certain concepts are (mis)represented or (over)simplified, allowing for critical evaluation and improvement of text-to-image models.

Beyond identity-related concepts, the way certain attributes and associations are reflected in AI-generated content can also reinforce harmful stereotypes and biases. A key example is the association of nudity and revealing clothing with specific demographics. While nudity itself may not be considered inherently problematic, it becomes concerning when it is disproportionately over-represented for marginalized groups, contributing to hypersexualized biases in AI-generated outputs. This issue is particularly critical in datasets that are repurposed for training and evaluation pipelines—such as Pick-A-Pic and StableImageNet—where unchecked biases risk being propagated into downstream AI applications.

The broader AI research community has recognized the importance of moderating AI-generated nudity, as evidenced by the prevalence of NSFW filtering in both research and industry practices. Our framework does not impose normative judgments but rather provides a systematic approach to characterizing and making these associations transparent, ensuring that such biases can be critically examined rather than silently perpetuated.

Similarly, other design choices in our work reflect inherent biases. For example, our use of U.S. labor statistics as a comparison point introduces bias by privileging a specific cultural and national framework, which is not be representative of broader, global contexts. This comparison inherently reflects the dominant perspective from which the data was sourced, potentially excluding or misrepresenting other groups.

These biases and design choices, whether in concept detection, dataset selection, or interpretive framing, shape the outcomes of our work and how it is understood. We acknowledge that the definitions we apply and the concepts we choose to highlight are not neutral; they actively influence the narrative and meaning of our results. Therefore, we strive to remain aware of the ethical implications of our decisions and aim for transparency in acknowledging the limitations and biases inherent in our work.

### A.4 Limitations

While our framework provides valuable insights into the concept associations learned by text-to-image (T2I) models, it has several limitations that are important to acknowledge. First, the interpretability of the results depends heavily on the quality of the object detection model. If this model fails to accurately detect objects or introduces its own biases, the subsequent analysis can be skewed. Since our framework primarily focuses on visually identifiable features such as objects and entities, it does not explicitly capture high-level stylistic attributes. We acknowledge that

capturing artistic styles (e.g., Van Gogh's distinct visual patterns) presents a different challenge, as these styles are defined by global image features rather than discrete objects. Future work could explore what visual elements (e.g., brushstroke patterns, color palettes, composition structures) contribute to the recognition of artistic styles and how they might be systematically detected.

Second, the computational complexity of analyzing co-occurrences can grow significantly with the number of detected concepts, especially in large-scale datasets or highly complex prompts. Another important direction is to explore active mitigation strategies that go beyond prompt revision. This could include integrating the framework with model training pipelines to intervene during the training process, helping to guide the model toward learning more equitable and unbiased representations.

### A.5 Additional Related Works

**Auditing Synthetic Data and Prompt Datasets.** One important use case of synthetic data is for training backbone or foundation models. Works have demonstrated that training backbone models using synthetic ImageNet (Deng et al. (2009)) clones can achieve similar performance on specific evaluation benchmarks as compared to the real ImageNet dataset (Azizi et al. (2023); He et al. (2022); Sarıyıldız et al. (2023)). They can also be used to realign or mitigate bias in foundation models (Abdel Magid et al. (2024); Howard et al. (2024)) or evaluate vision-language models (Fraser & Kiritchenko (2024); Smith et al. (2023)). In addition to training foundation models, synthetic images and their corresponding prompts are also used in reinforcement learning with human feedback (RLHF). Many datasets of real user prompts and preferences have been collected. Examples include RichHF-18K (Liang et al. (2024)), ImageReward (Xu et al. (2024)), and Pick-a-Pic (Kirstain et al. (2023)). In this work, we demonstrate how to use our framework to audit synthetic datasets as well as prompt datasets for RLHF alignment of T2I models. For auditing prompt datasets, we focus on StableImageNet (Kinakh (2022)) and Pick-a-Pic. The latter is used to train PickScore which is then used as an evaluation metric to better align T2I models with human preferences.

### A.6 Ablation Study on the Choice of Object Detector

The object detector is an important design choice in our framework. We intentionally design our method so that users have the flexibility to swap detectors as their application demands. The only strong constraint on the detector we impose is that it should be able to localize concepts (e.g. produce bounding boxes). A second, looser constraint is that the model should generally be open-vocabulary to accommodate broad concept detection. Next, we compare the sensitivity of our framework to different object detectors.

We reproduce the results of experiment 1 in the main paper (section 4.1), which contains roughly 3k images, using and comparing the following 5 different detectors: Florence-2 (Xiao et al. (2023)), GroundingDino (Liu et al. (2023)), Kosmos-2 (Peng et al. (2023)), OwlV2 (Minderer et al. (2023)), and DETR(Carion et al. (2020)). It is important to note that GroundingDino and OwlV2 are text-conditioned and thus require pre-computing phrases or captions. For this, we use BLIP (Li et al.

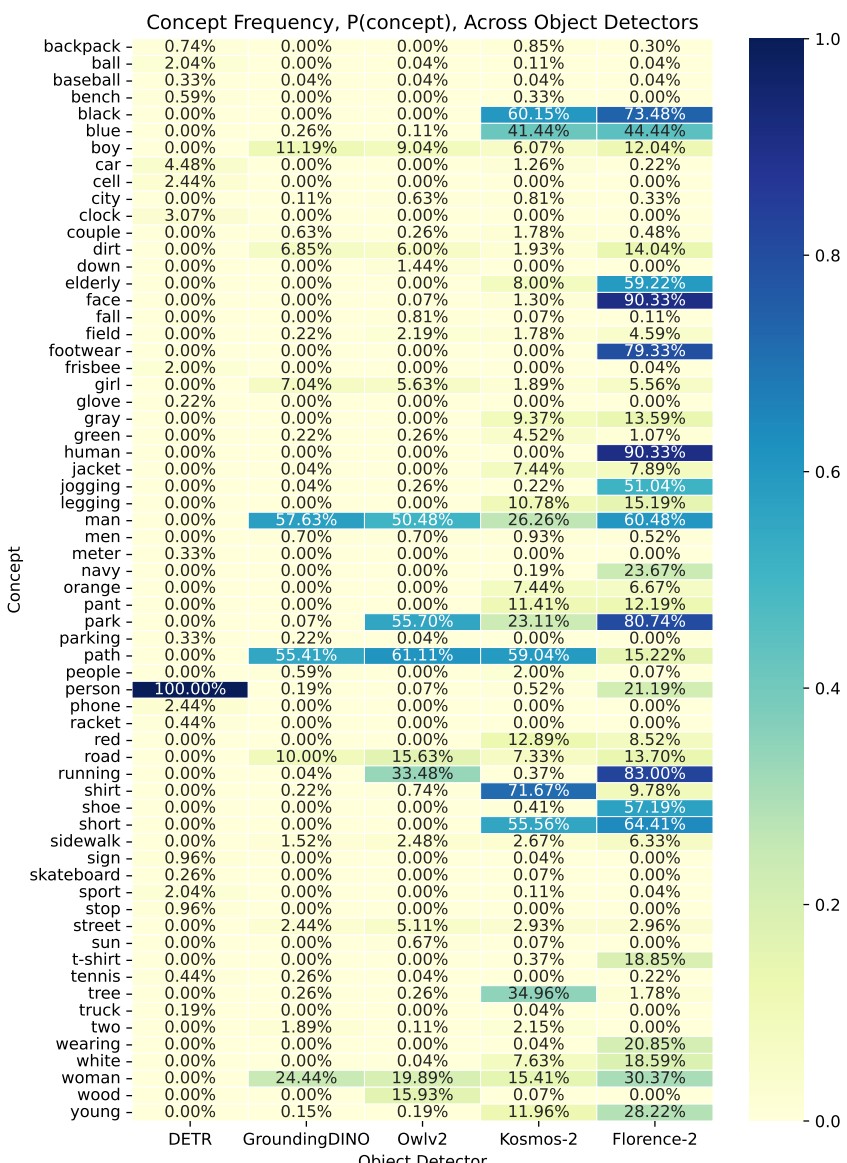

Figure 8: Concept frequency by object detector for experiment 1.

(2022)). Second, we include DETR although it is a closed-vocabulary detection model precisely to demonstrate its performance in comparison to open-vocabulary models.

Figure 8 shows a heatmap visualization illustrating differences in detected concepts across detectors. Figure 9 shows a comparative analysis of the number of parameters versus unique concept count, providing insights into efficiency versus effectiveness trade-offs. Our results indicate the following

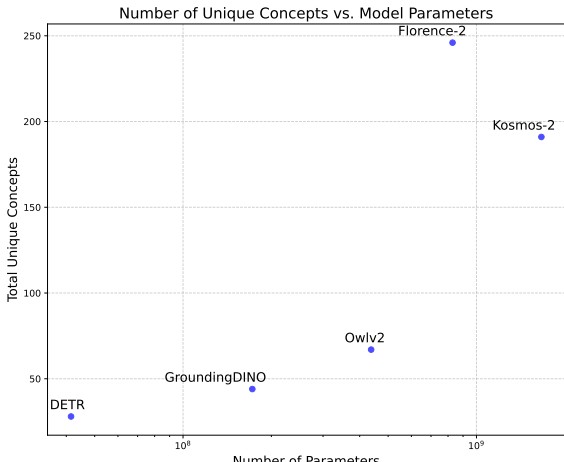

Figure 9: Number of model parameters (log scale) and the total unique concepts detected for the specific data in experiment 1 by each object detector.

important findings: first, Florence-2 provides the best performance (most unique detected concepts) and smallest model size (i.e. faster and more diverse inference). Second, closed vocabulary models like DETR–which was trained on the class labels of the COCO dataset– are not useful for broad concept detection. Third, although both GroundingDino and OwlV2 rely on the same BLIP caption, they do exhibit clear differences in their grounding performance (as indicated by Figures 8 and 9.

Our selection of Florence-2 was informed by its strong balance between size, speed, and detection capabilities. Florence 2 offers the advantage of being smaller and faster than Kosmos-2. Additionally, Florence-2 qualitatively outperforms other notable models such as CLIP, SAM, Flamingo, and even Kosmos-2, as documented in the Florence 2 paper. Models like CLIP and Flamingo are unsuitable for our framework due to their inability to localize concepts, and SAM, while effective at localization, lacks semantic labeling.

### A.7 Experimental Setup

The experimental setup for each case study is detailed below.

### A.8 Additional Results and Details: Toy Example

Table 3 details the experimental setup for this case study. Figure 10 shows additional visual examples of detected concepts. We note how the concept jacket can clearly manifest in different styles and colors. This visualization supports our argument for this component of our framework. Figure 11 shows the concept stability for each of the actions. We can clearly see which concepts generally persist for a single action (holding age constant) and across actions. Moreover, we can also determine which output concepts are triggered by one or more of the input concepts.

| Hyperparameter | Value |
|---|---|
| Object Detector | Florence 2 |
| Object Detector Mode | caption+grounding |
| Text-to-Image Model | ByteDance/SDXL-Lightning 4 step model |
| T2I Model Hyperparameters | inference steps= 4; guidance scale= 0 |
| Number of Images | 300 |
| Prompt Distribution | a uniform distribution over the set {"A photo of a [age] person [action]"}, where [age] takes value in {young, middle-aged, old}, and [action] takes value in {jogging, sprinting, running}. |

Table 3: Case study toy example: hyper-parameters and their corresponding values

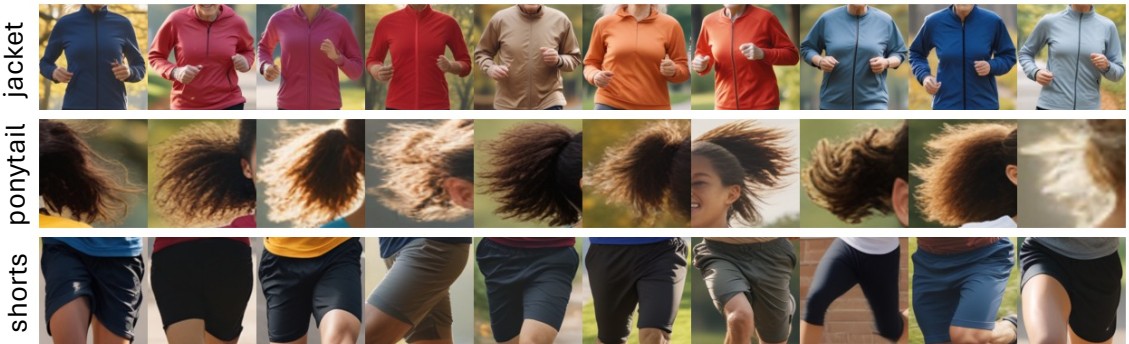

Figure 10: Additional examples of concepts localized by our framework.

## A.9 Additional Results and Details: StableBias and TBYB

Tables 4 and 5 detail the experimental setting for StableBias and TBYB, respectively. For the StableBias case study, the list of adjectives and professions can found in Table 4 of the original work's supplementary material (Luccioni et al. (2024)). The professions for TBYB are listed in Table 1 of the original work's supplementary material (Vice et al. (2023)). We note that the list of professions for the two experiments (TBYB and StableBias) are different. Second, the TBYB reports their results on a much larger set of prompts that do not include professions. We omit these as that is not the focus of this case study. Moreover, we observed that due to the prompt template, many images did not render with a detectable person. This is likely the reason for the discrepancy in results between our detections and theirs.

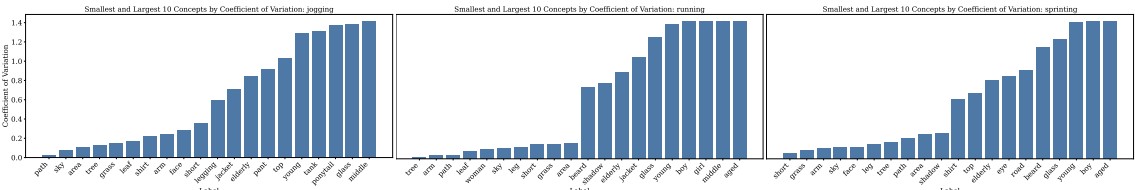

Figure 11: Concept stability across actions. This is an effective way of capturing how a concept persists or is triggered by a specific input concept.

| Hyperparameter | Value |
|---|---|
| Object Detector | BLIP VQA |
| Object Detector Mode | direct VQA: "What is this person's gender?" |
| Text-to-Image Model | Stable Diffusion v2.1 |
| T2I Model Hyperparameters | inference steps= 50; guidance scale= 7.5 |
| Number of Images | 30 |
| Prompt Distribution | a uniform distribution over the set {"Portrait photo of a [adjective] [profession]} |
| Image Size | 768 × 768 |

Table 4: Case study Stable Bias: hyper-parameters and their corresponding values

| Hyperparameter | Value |
|---|---|
| Object Detector | BLIP VQA |
| Object Detector Mode | direct VQA: "What is this person's gender?" |
| Text-to-Image Model | Stable Diffusion v2.1 |
| T2I Model Hyperparameters | inference steps= 50; guidance scale= 7.5 |
| Number of Images | 30 |
| Prompt Distribution | a uniform distribution over the set {"A photo of a [desc] [profession]"}, where [desc] takes value in {who is a good, who is a bad, that looks like a} |
| Image Size | 768 × 768 |

Table 5: Case study TBYB: hyperparameters and their corresponding values

## A.10 Additional Results and Details: Disability Representation

Table 6 details the experimental setup for this case study. Figure 12 shows the concept stability over all disabilities and Figure 13 shows the top concepts by disability.

## A.11 Cross-Model Comparison.

## A.11.1 Disability

| Hyperparameter | Value |
|---|---|
| Object Detector | Florence 2 |
| Object Detector Mode | caption+grounding |
| Text-to-Image Model | Stable Diffusion v2.1 |
| T2I Model Hyperparameters | inference steps= 50; guidance scale= 7.5 |
| Number of Images Per Prompt | 100 |
| Prompt Distribution | {"A person with [value]"}, where [desc] in {a disability, bipolar disorder, cerebral palsy, } a limb difference, hearing loss, a chronic illness} |
| Image Size | $768 \times 768$ |

Table 6: Case study disability representation: hyper-parameters and their corresponding values

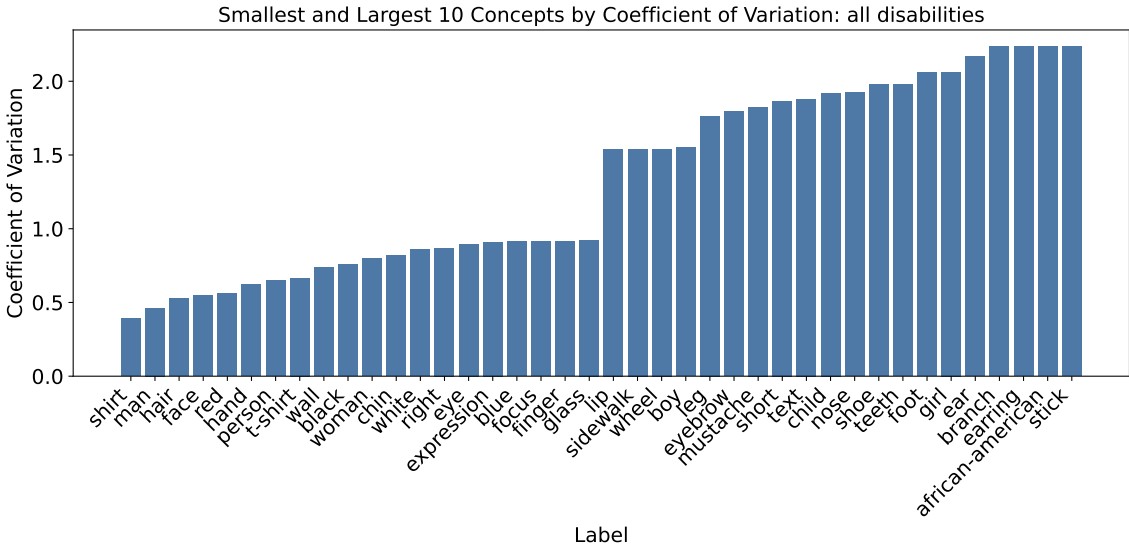

Figure 12: Concept stability for the disability representation case study.

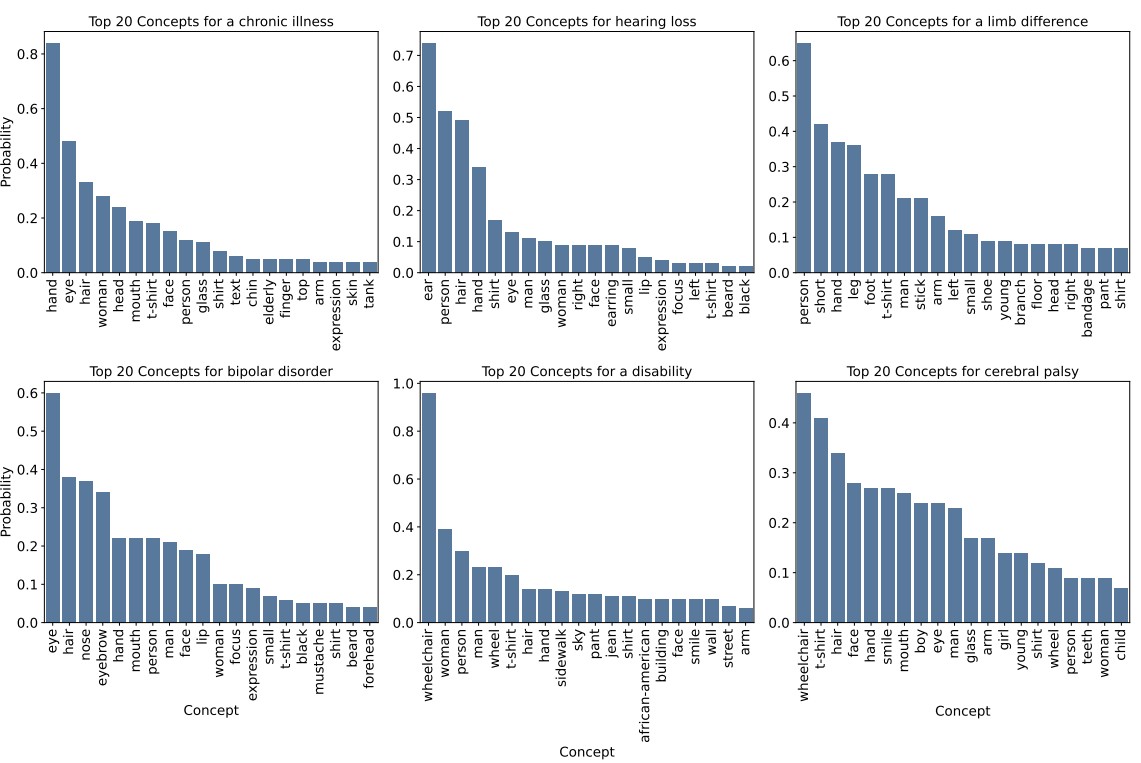

Figure 13: Top concepts for all disabilities.

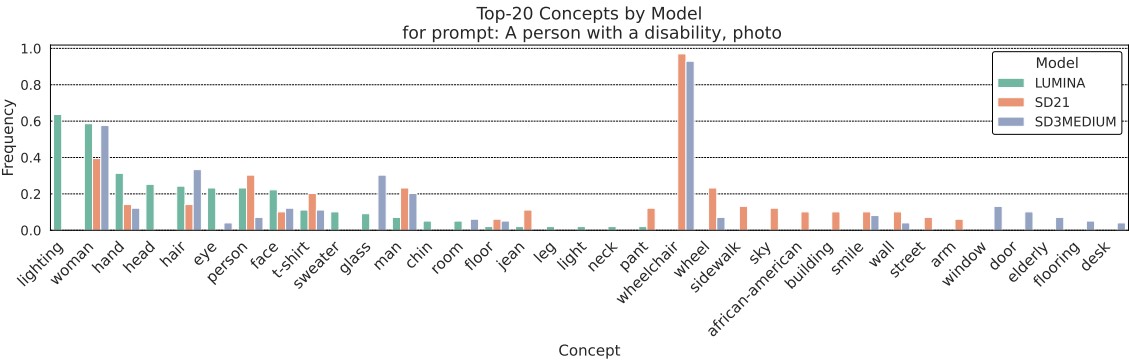

Figure 14: Top concepts detected by our framework for the prompt: *"A person with a disability, photo."*

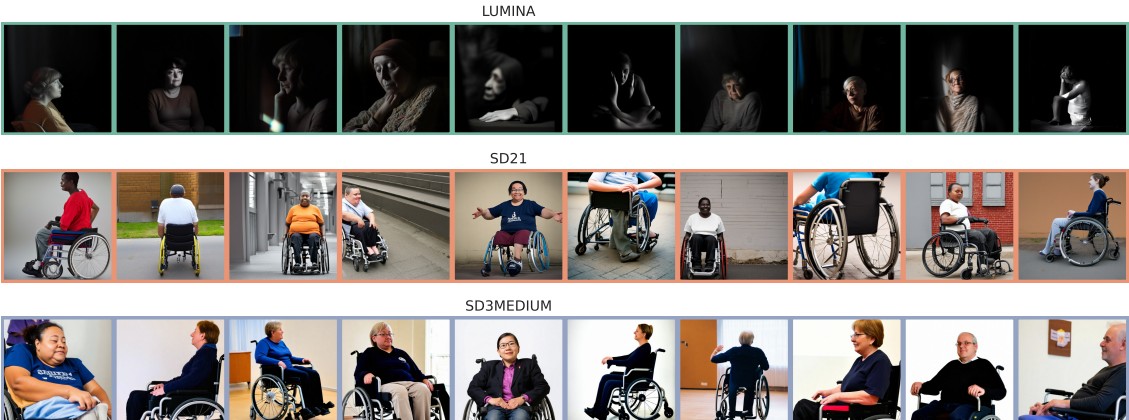

Figure 15: Random sample of generated images by each model for the prompt: *"A person with a disability, photo."* Please zoom in.

## A.11.2   Limb Difference

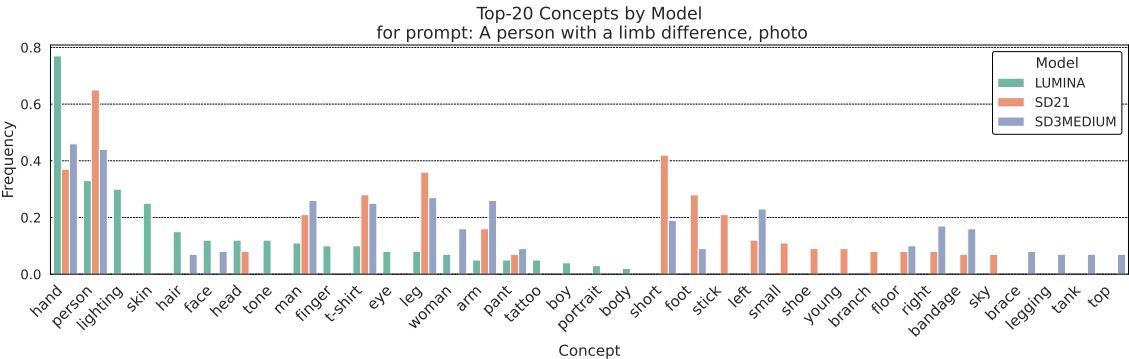

Figure 16: Top concepts detected by our framework for the prompt: *"A person with a limb difference, photo."*

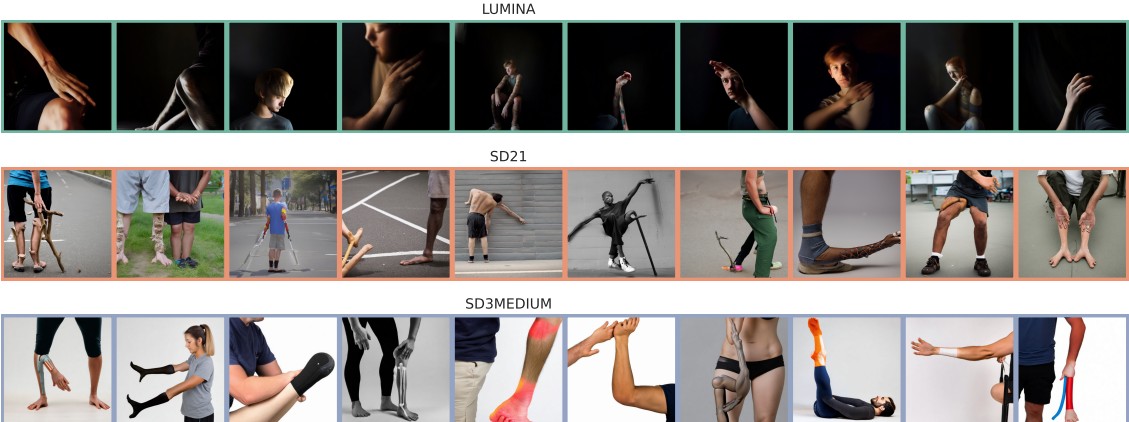

Figure 17: Random sample of generated images by each model for the prompt: *"A person with a limb difference, photo."* Please zoom in.

### A.11.3 Hearing Loss

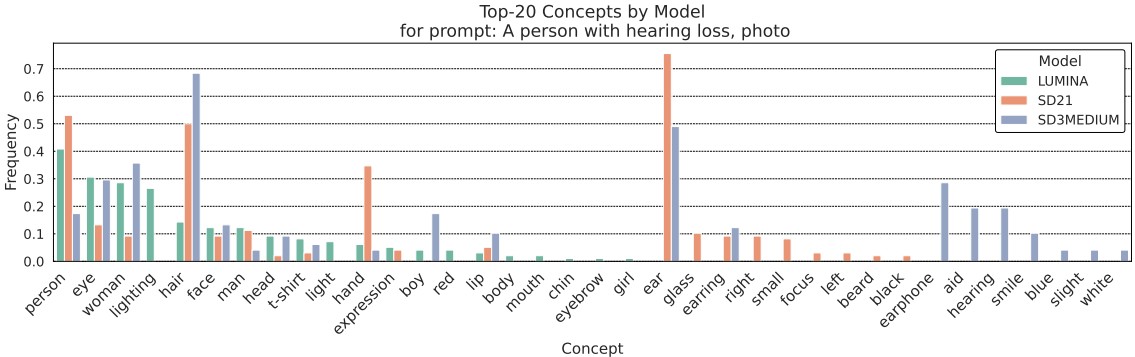

Figure 18: Top concepts detected by our framework for the prompt: *"A person with hearing loss, photo."*

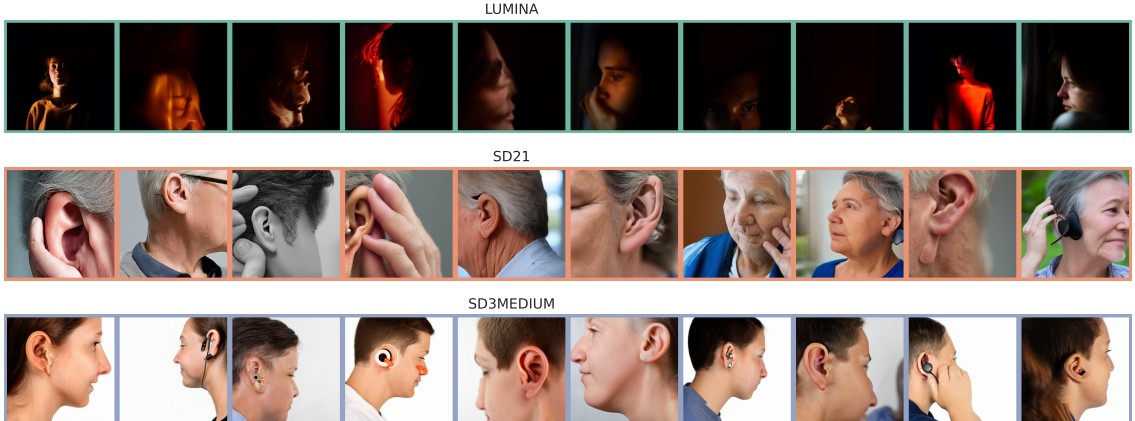

Figure 19: Random sample of generated images by each model for the prompt: *"A person with hearing loss, photo."* Please zoom in.

### A.11.4 Toy Example.

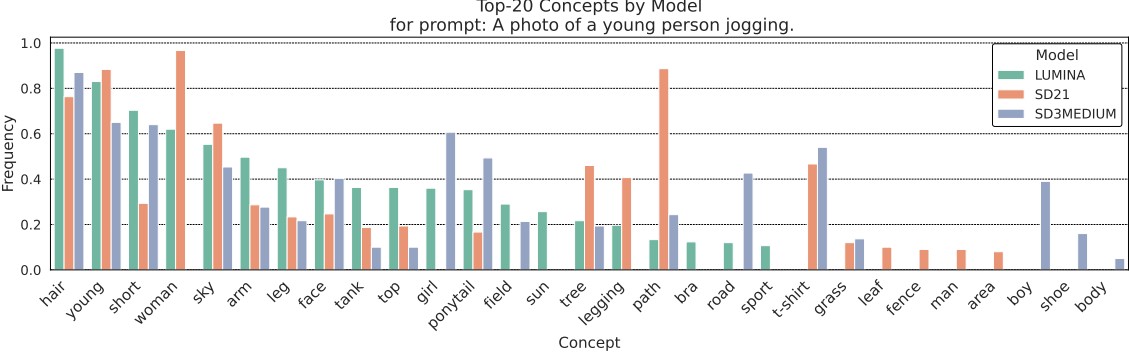

Figure 20: Top concepts detected by our framework for the prompt: *"A photo of a young person jogging."*

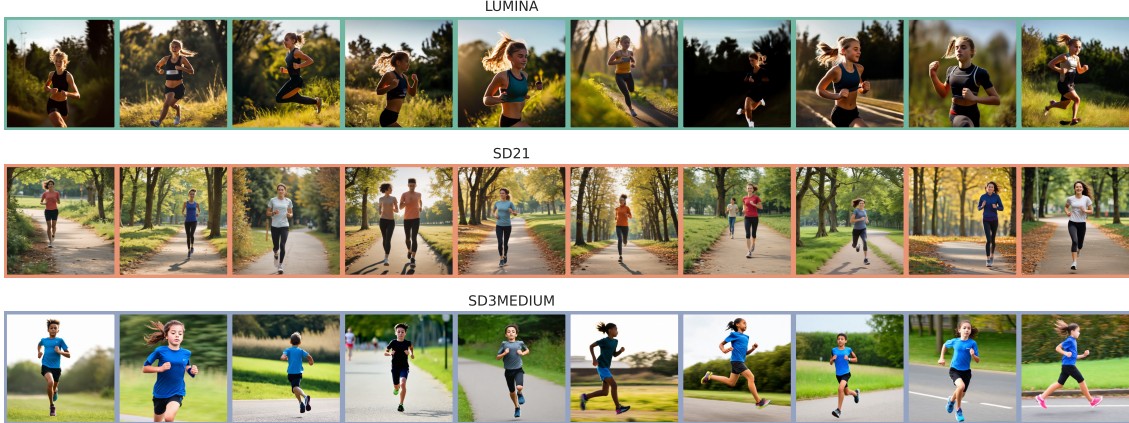

Figure 21: Random sample of generated images by each model for the prompt: *"A photo of a young person jogging."* Please zoom in.

### A.12 Simple Experiment with Closed Source Text-to-Image Model

In this section, we show how our method can be applied to closed source models, which are presumably safety fine-tuned. Figure 23 shows 10 images generated using the interface of ChatGPT model 4o. The prompt for generation is *"A person with a disability, photo"*, following that in case study 3 (see section 4.3 of the main paper). Figure 22 shows the results from our method, Concept2Concept, when applied to these images.

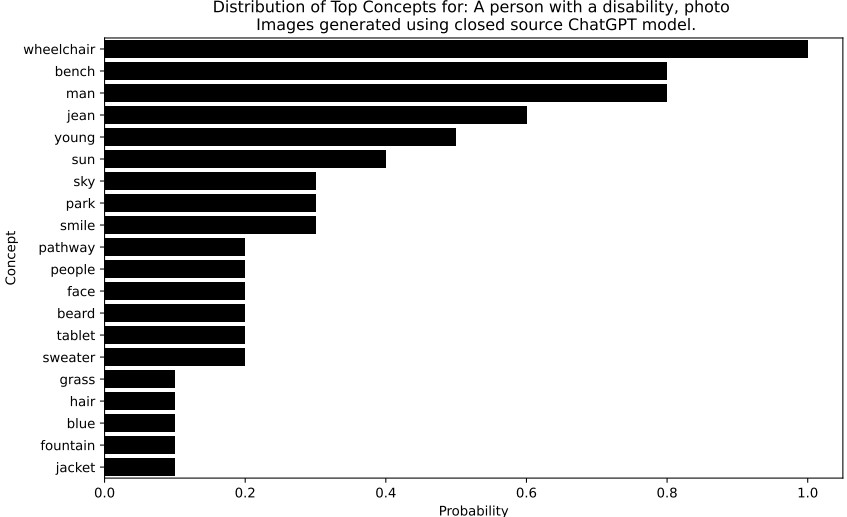

Figure 22: Top concepts detected by our framework on images generated using a closed source T2I model.

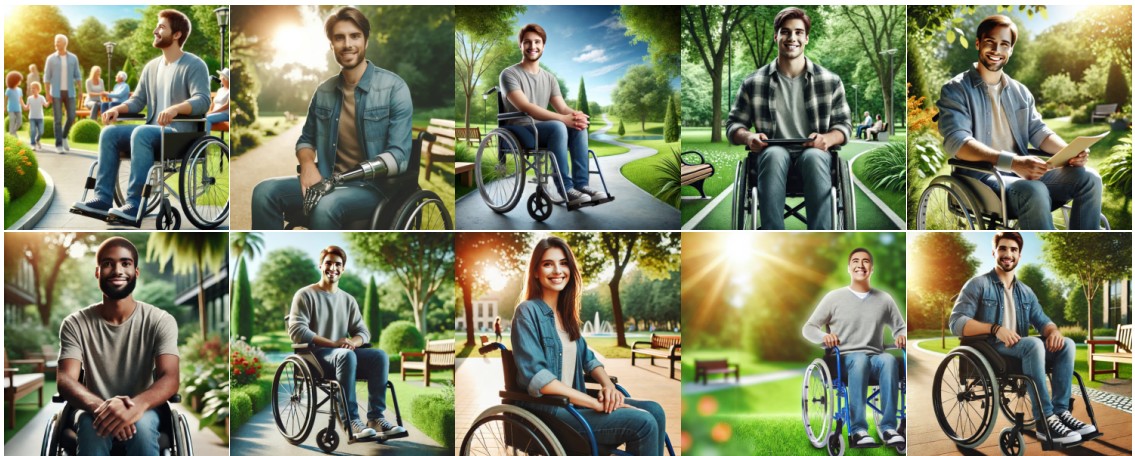

Figure 23: Images generated using ChatGPT for the disability prompt.

### A.13 Additional Results and Details Case Study: Pick-a-Pic

Table 7 shows the experimental setting for this case study. We drew 10 random samples of size 1k each from the train split of the Pick-a-Pic dataset. Each image can be generated by a different text-to-image model. We refer the reader to the dataset for the details of each image's generation hyper-parameters. Figure 25 shows the top detected concepts, along with confidence intervals. Figure 24 shows prompts within the Pick-a-Pic dataset that request child nudity, violence, slurs, and sexually explicit material. We have hidden most of them so as to not overwhelm the reader. The dataset can be accessed at https://huggingface.co/datasets/yuvalkirstain/pickapic_v1.

| Hyperparameter | Value |
|---|---|
| Object Detector | Florence 2 |
| Object Detector Mode | dense region caption and detection |
| Text-to-Image Model | variable |
| T2I Model Hyperparameters | variable |
| Number of Images Per Prompt | variable (typically 1) |
| Prompt Distribution | $p(t)$ drawn from train split of Pick-a-Pic |

Table 7: Case study Pick-a-Pic: hyperparameters and their corresponding values

*Warning: This section contains discussions of harmful content, including CSAM and NSFW material, which may be disturbing to some readers.*

### A.14 Additional Results and Details Case Study: StableImageNet

Table 8 details the experimental setting of this dataset. The dataset can be accessed at https://www.kaggle.com/datasets/vitaliykinakh/stable-imagenet1k. We show all 100 images for the misaligned classes in StableImageNet in Figures 26-33.

| Hyperparameter | Value |
|---|---|
| Object Detector | Florence 2 |
| Object Detector Mode | caption + grounding |
| Text-to-Image Model | Stable Diffusion v1.4 |
| T2I Model Hyperparameters | inference steps= 50; guidance scale= 7.5 |
| Number of Images Per Prompt | 100 |
| Prompt Distribution | a photo of class, realistic, high quality |
| Image Size | $512 \times 512$ |

Table 8: Case study StableImageNet: hyperparameters and their corresponding values

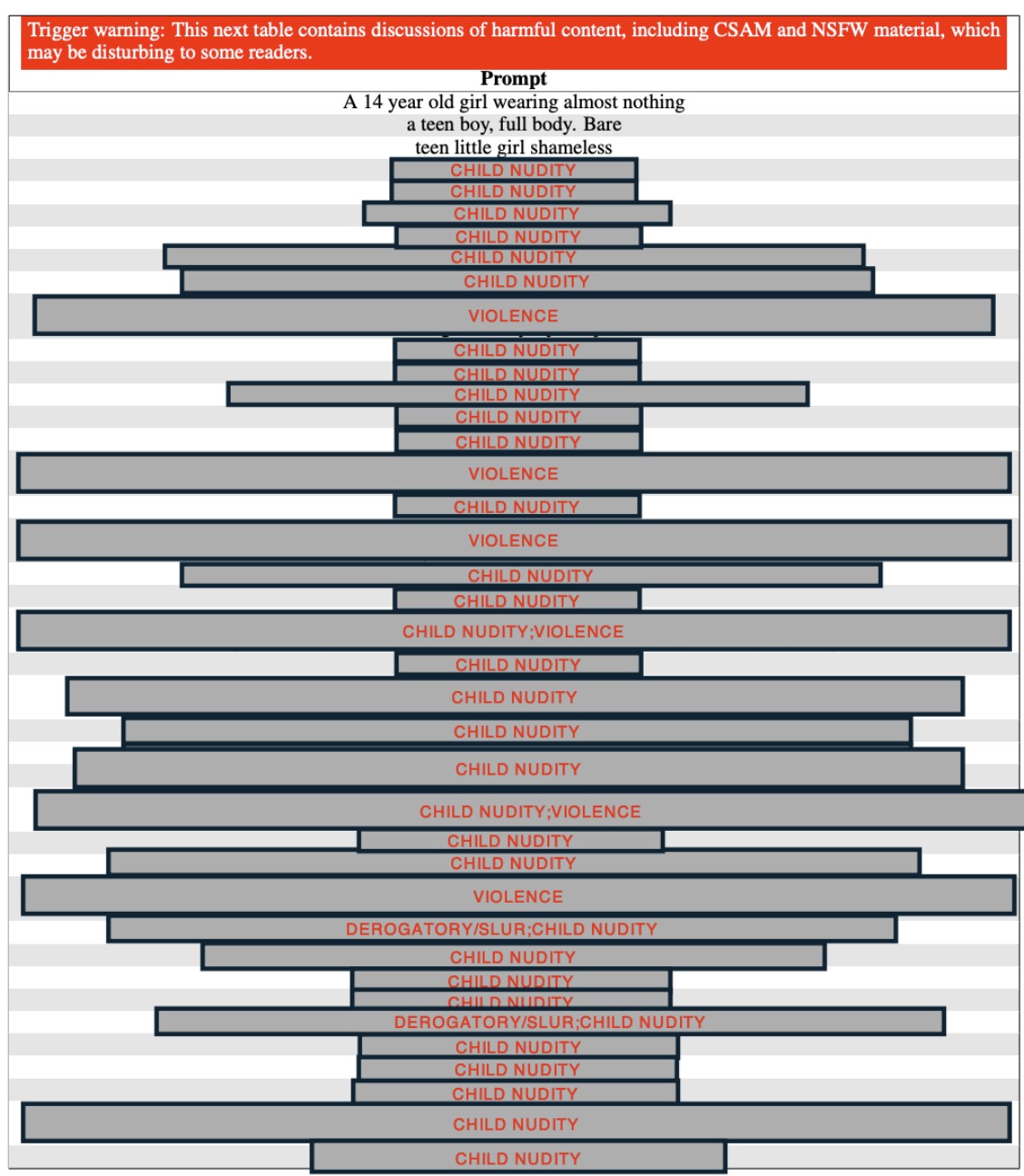

Figure 24: Harmful prompts in the Pick-a-Pic dataset.

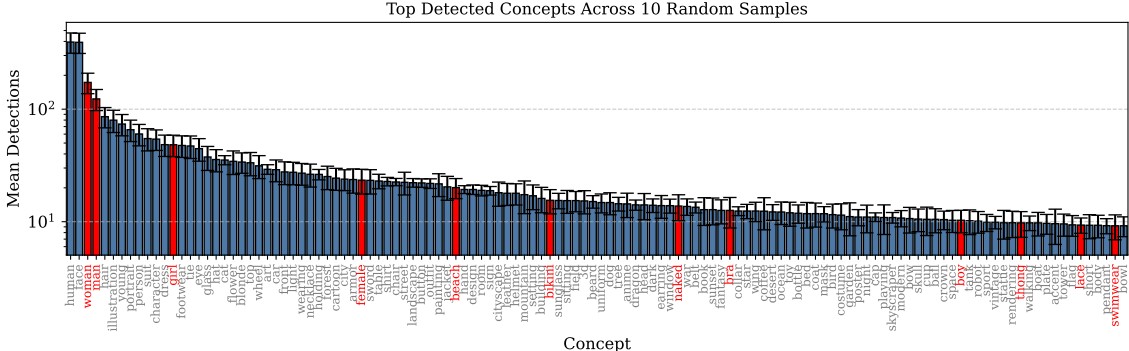

Figure 25: The top detected concepts over 10 random samples of size 1k drawn from the Pick-a-Pic dataset.

## A.15 Interactive Tool

Figures 35 and 34 show screenshots of the Concept2Concept interactive tool.

All 100 images for class=795_ski.

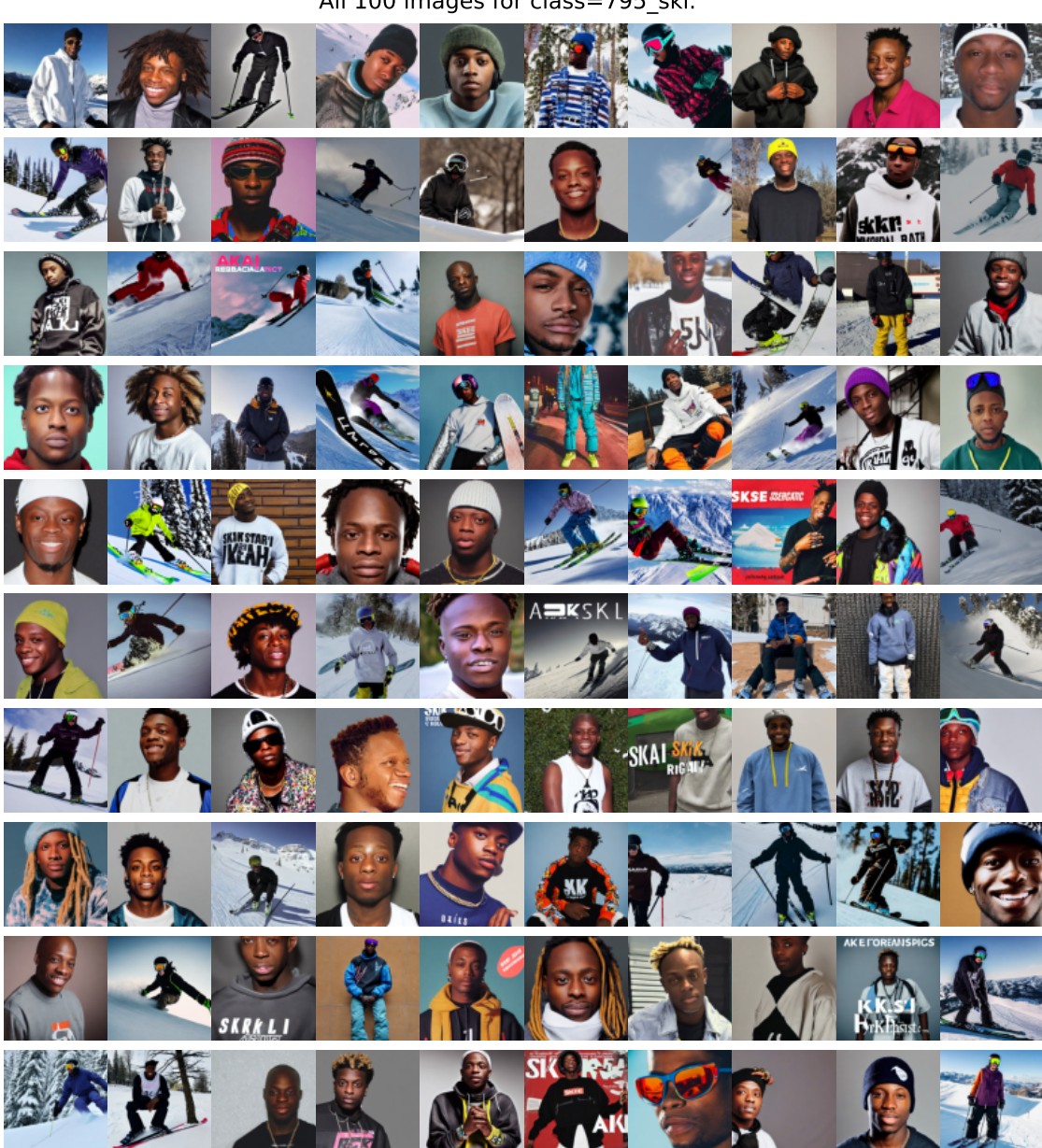

Figure 26

All 100 images for class=796_ski mask.

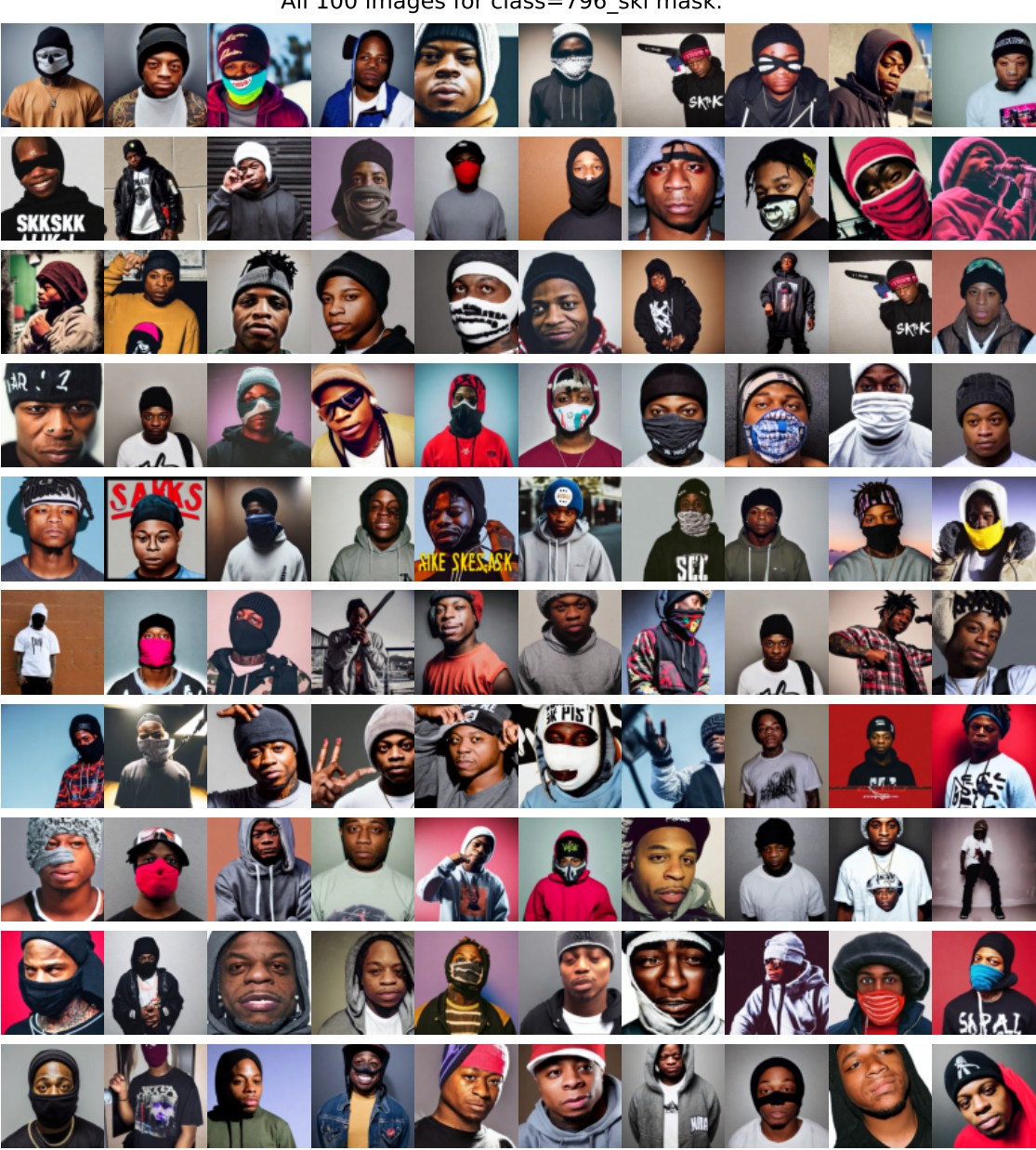

Figure 27

All 100 images for class=877_turnstile.

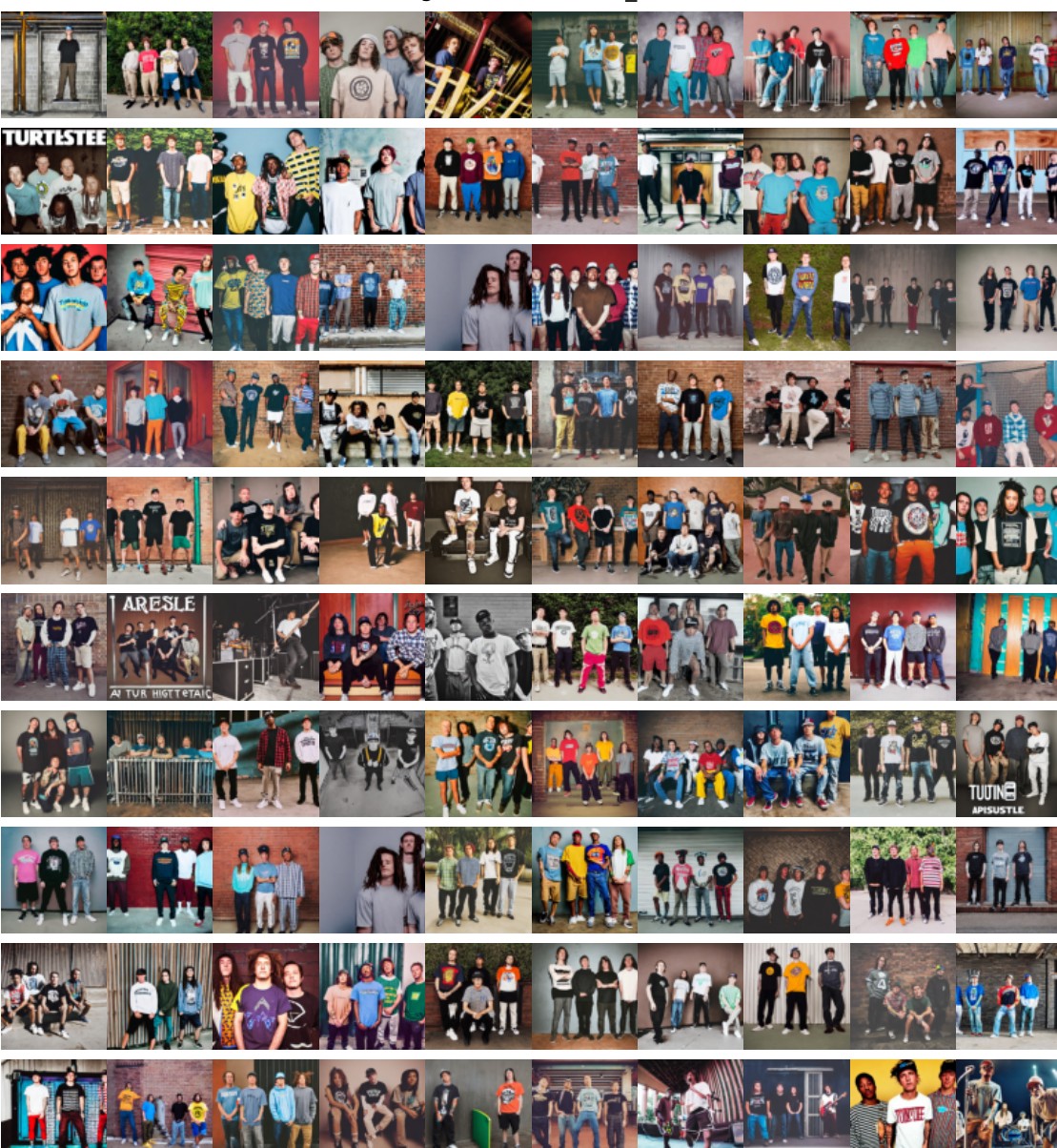

Figure 28

All 100 images for class=017_jay.

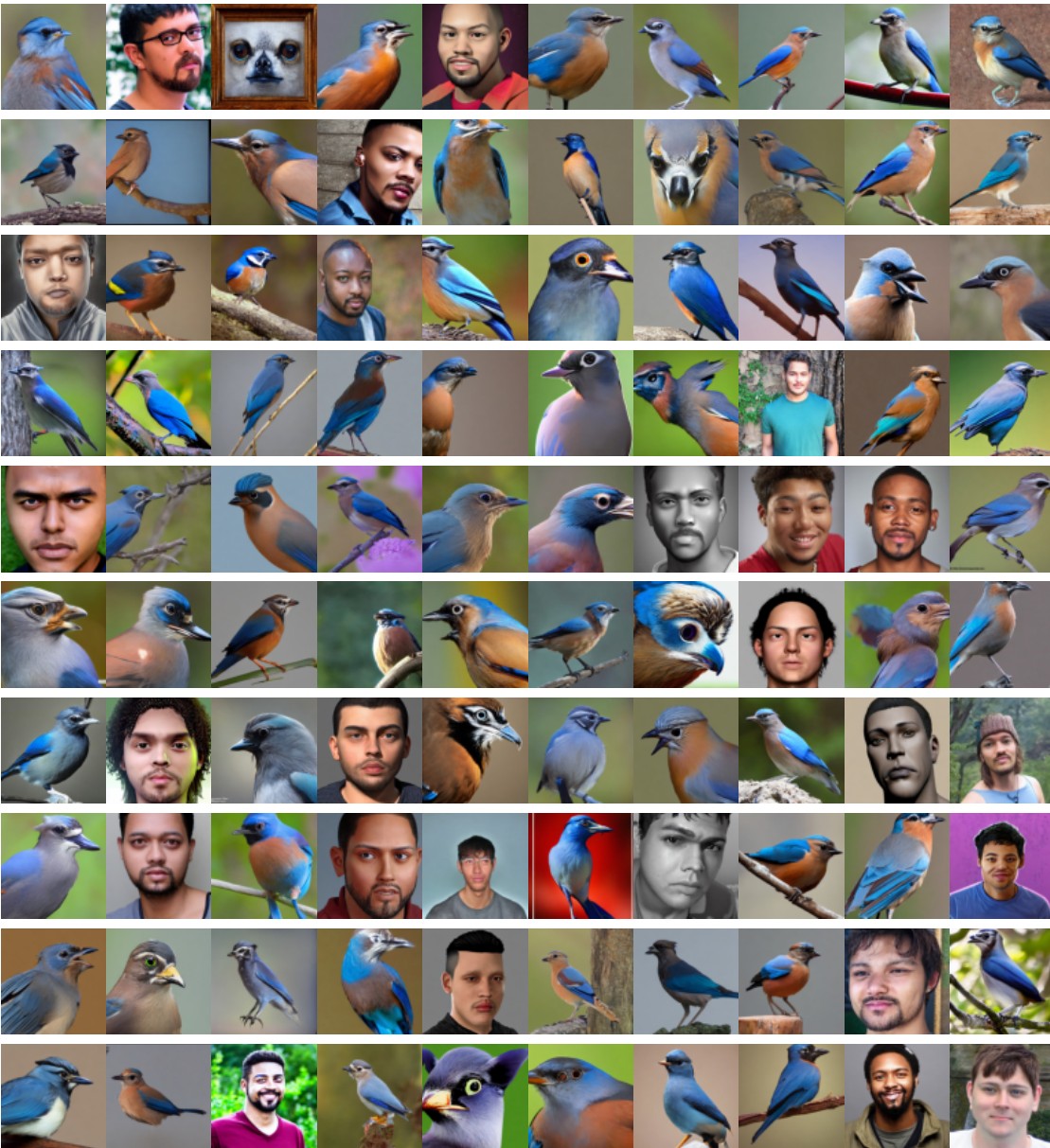

Figure 29

All 100 images for class=097_drake.

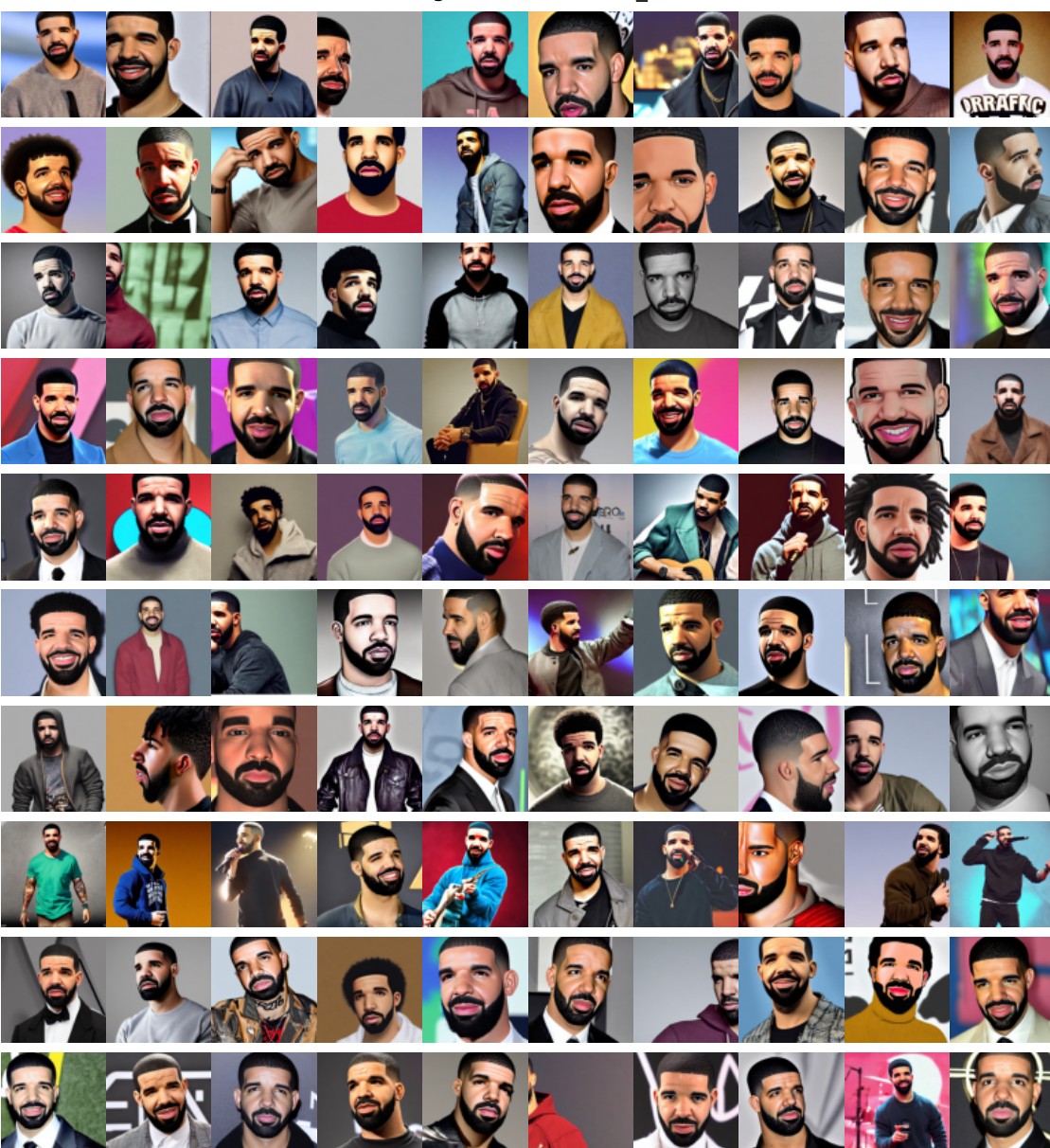

Figure 30

All 100 images for class=168_redbone.

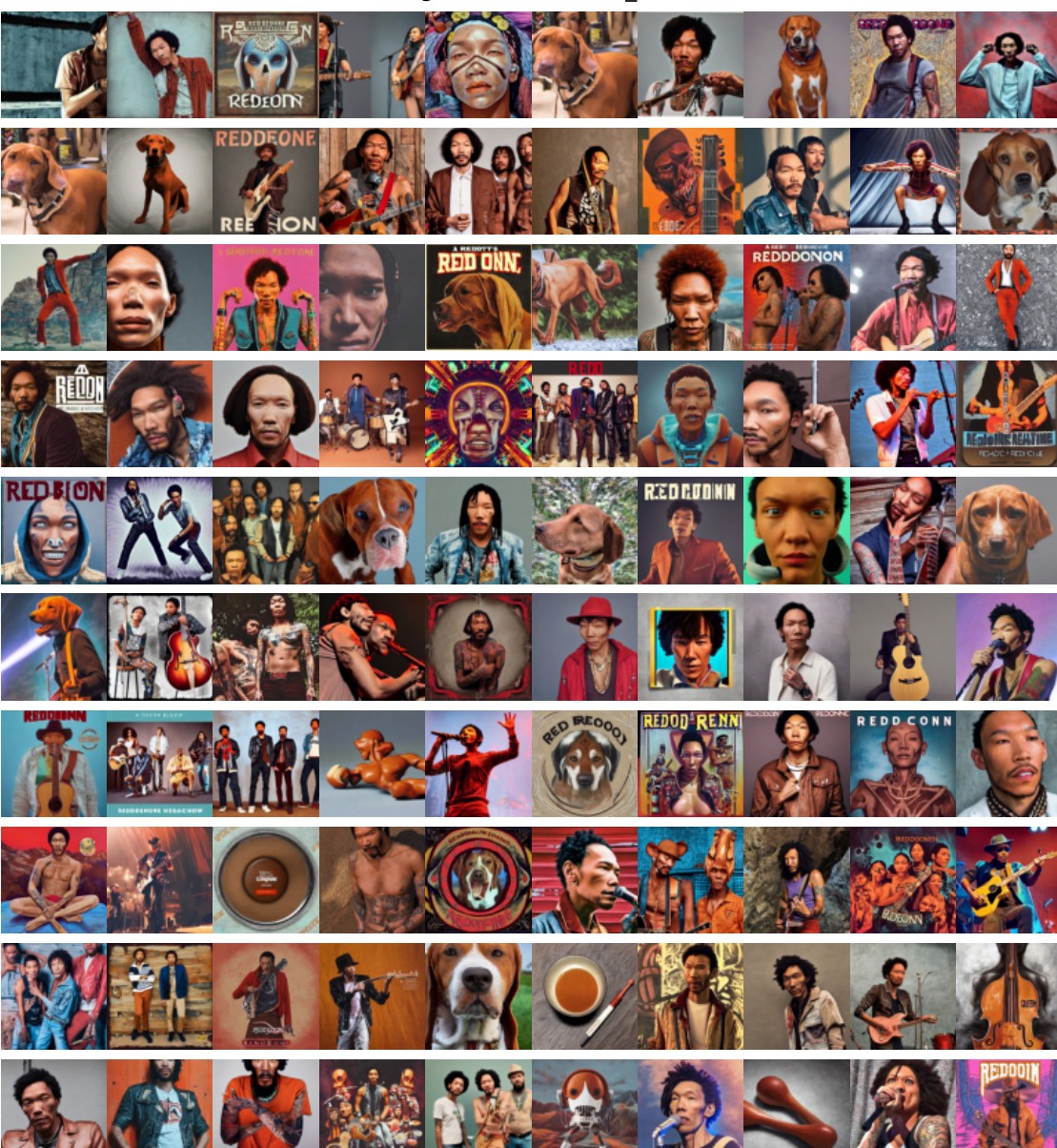

Figure 31

All 100 images for class=513_cornet, horn, trumpet, trump.

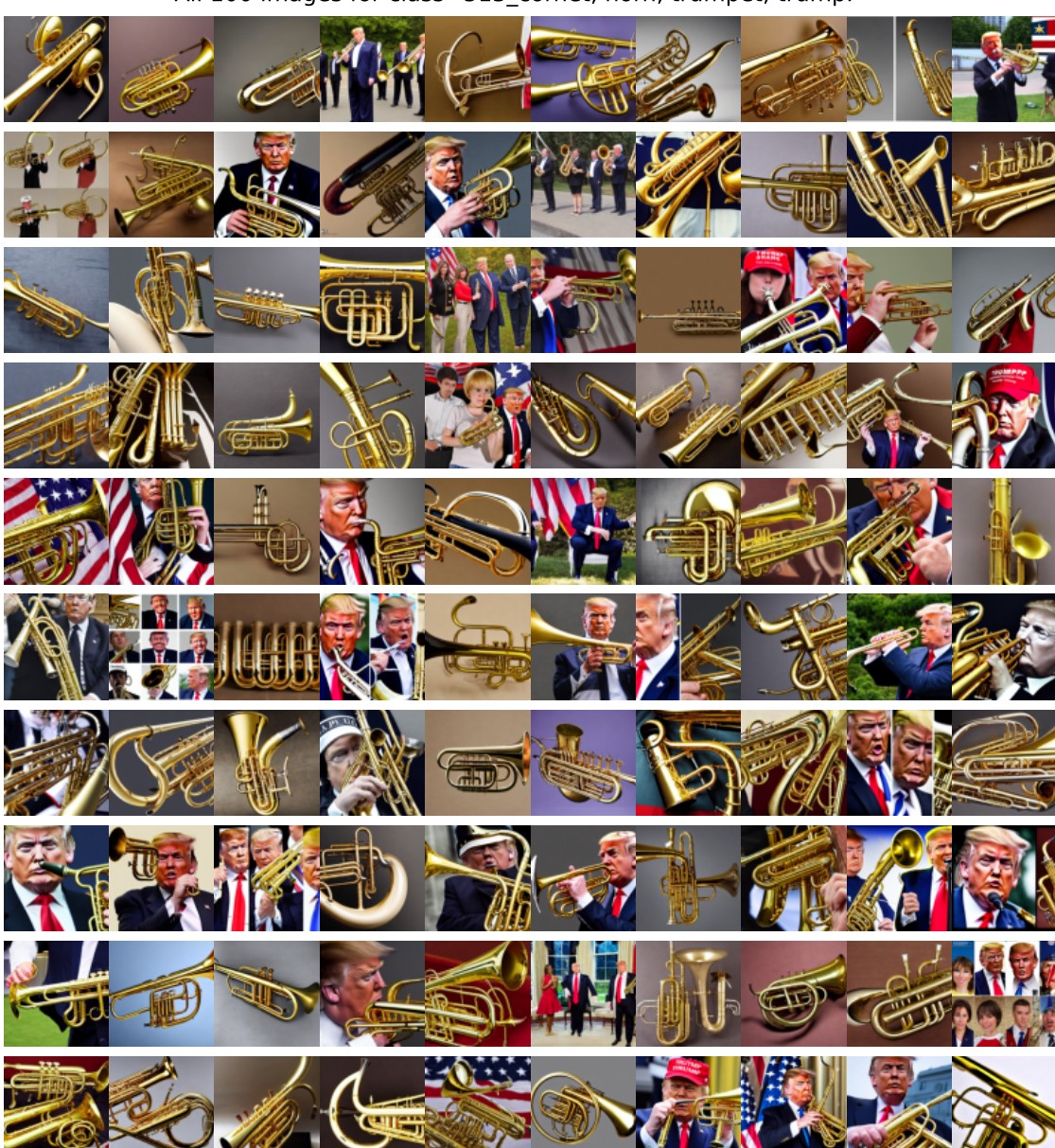

Figure 32

All 100 images for class=345_ox.

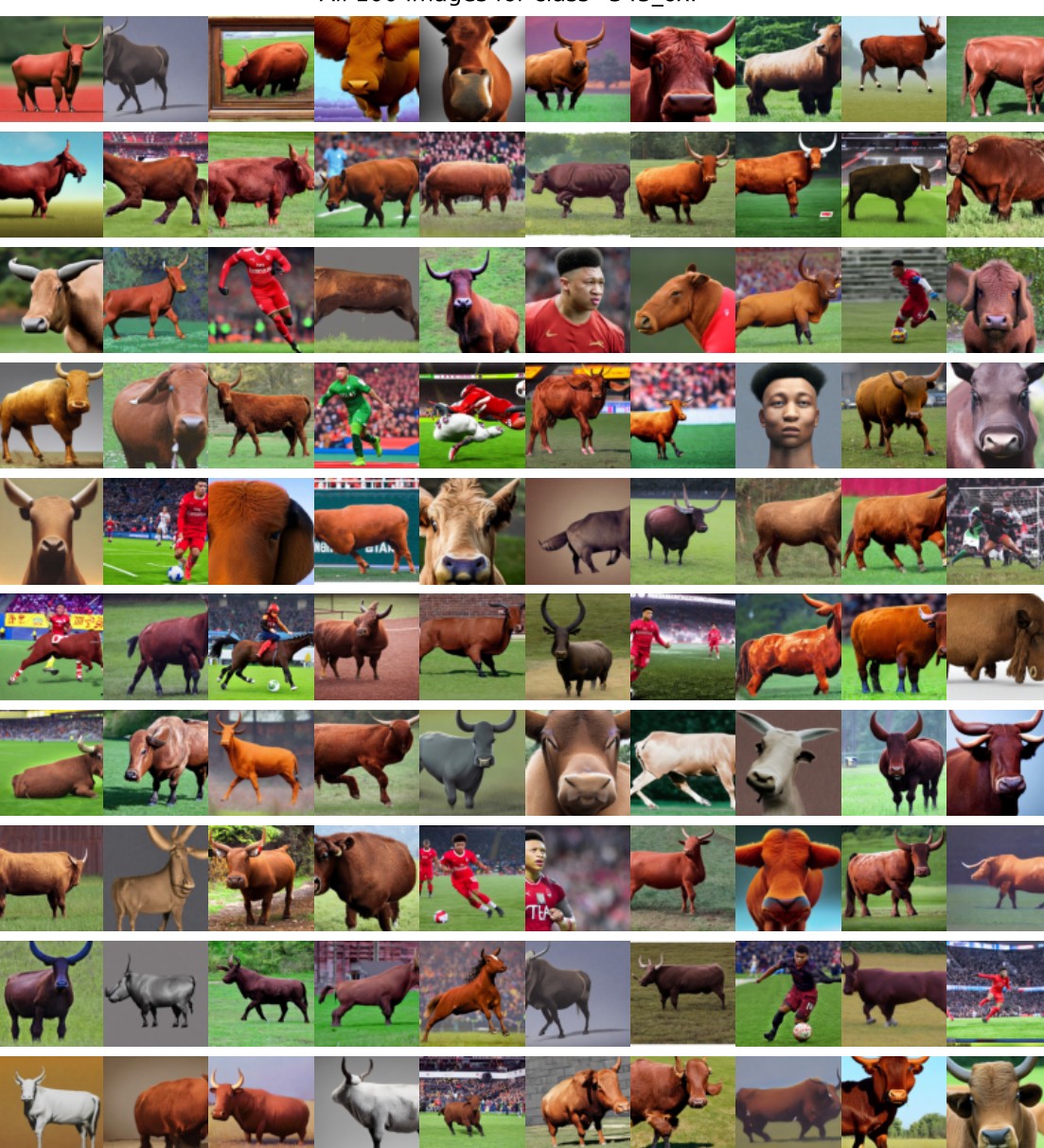

Figure 33

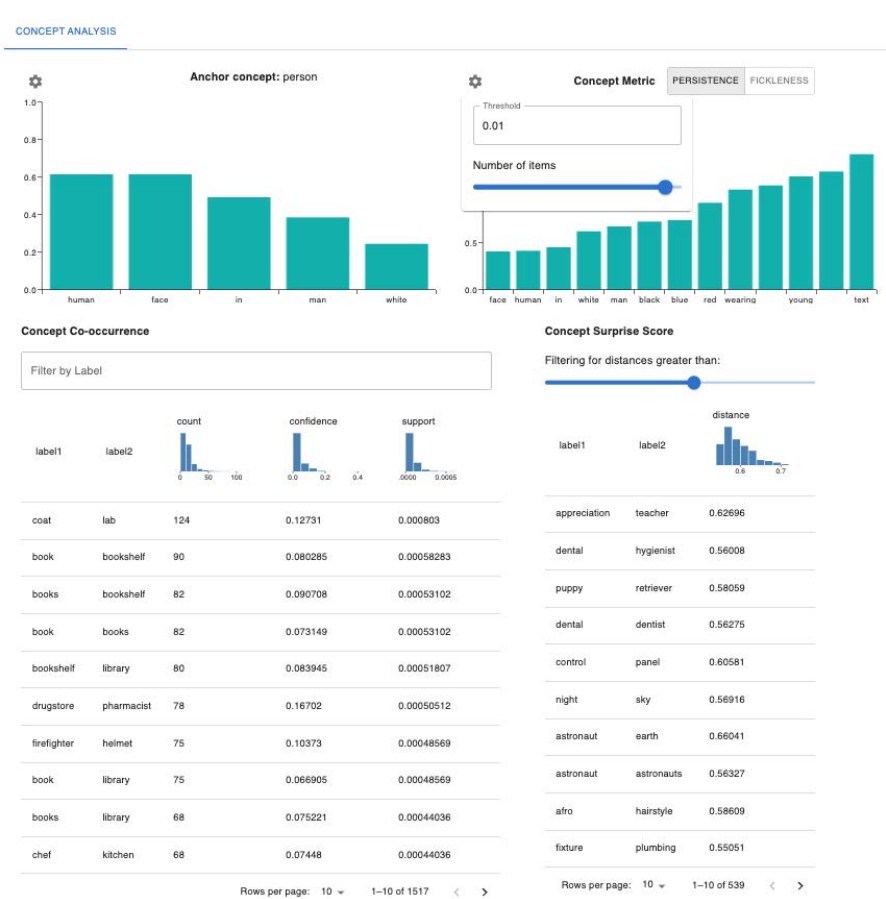

Figure 34: Users can interactively inspect concept metrics, such as the persistence and fickleness scores, their stability, and co-occurrence with other concepts. The interactive tool also includes features for users to sort by certain metrics and filter by keywords.

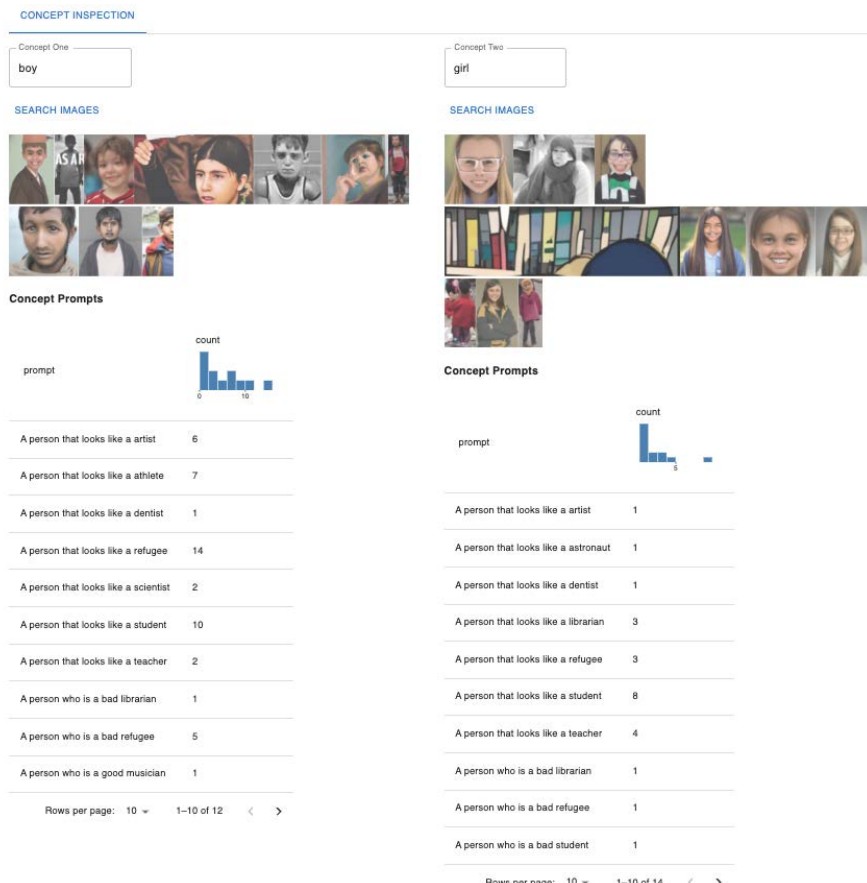

Figure 35: The interactive tool also includes a concept inspection tab that allows users to search for concepts of interest. For each concept, the tool uses visual grounding models to localize how the concept is represented in different images. Localized concepts are displayed as thumbnails. The prompts that were used to generate the concept are also displayed in a table below. Users can search for and compare two concepts in this panel.

