# OpenReview forum: "Is What You Ask For What You Get? Investigating Concept Associations in Text-to-Image Models"
_TMLR — Accepted by TMLR_

### Review · Reviewer_fziz · 2025-02-22

**Summary Of Contributions:**

The paper introduces Concept2Concept, a framework that audits T2I models by extracting and analyzing high-level concepts from generated images, enabling interpretable evaluations of how prompts influence outputs. Using metrics like concept frequency, stability, and co-occurrence, the framework uncovers biases and unexpected associations, such as gender stereotypes, misrepresentations of disabilities, and inappropriate content in training datasets like Pick-a-Pic. It also provides an open-source interactive tool that allows users to explore concept associations, supporting more transparent and ethical AI development.

**Audience:**

Yes

**Broader Impact Concerns:**

- Potential for misuse: The framework could be misused to identify and exploit biases in T2I models, enabling malicious actors to generate harmful content more effectively.
- Limited consideration of cultural context: The paperfocuses on Western cultural norms and datasets, limiting its applicability to other regions and demographics. Concepts like "nudity" or "disability" may be interpreted differently across cultures, yet the framework applies a one-size-fits-all approach.

**Claims And Evidence:**

No

**Requested Changes:**

1. Use multiple object detectors and compare their performance to improve reliability.
2. Setup comprehensive and systematic studies instead of case studies.

**Strengths And Weaknesses:**

Strengths
+ The paper provides a general framework to analyze biases in any forms as users desire.
+ The paper discovers and reproduces some biases existing in current T2I models regarding gender, age, etc.
+ The authors open-source the framework.

Weaknesses

- Concept associations are analyzed without considering the broader context of the image or prompt. For example, detecting a "thong" or "naked" might be appropriate in some artistic or cultural contexts but is flagged as problematic.
- The analysis relies heavily on the object detection models. Althoug the authors have mentioned in the Limitations section, they do not specify clearly how to mitigate the influence. The authors could report human evaluation on (a small portion of) images analyzed in this study. Additionally, the choice of detectors needs to be well-justified, and, adding comparison with alternative detectors will be good.
- The detectors may not work well on non-photorealistic images, such as cartoon, abstract, surreal, or artistic images.
- Some experiments lack generalizability. For example, section 4.1 studies only three actions on one model (SD2.1). Readers are not sure whether the conclusion can be generalized to a broader range of concepts and models.
- The framework could be misused to identify and exploit biases in T2I models for malicious purposes, such as generating harmful content more effectively.

---

> ### Author Response · Authors · 2025-03-07
> **Individual Comments**
>
> Thank you for your feedback. We have addressed your other points in the shared responses above. Below we address your comments that were not shared by other reviewers:
>
> **The framework could be misused to identify and exploit biases in T2I models for malicious purposes, such as generating harmful content more effectively.**
>
> We appreciate your thoughtful consideration of potential misuse scenarios. However, we clarify that our framework inherently focuses on identifying and visualizing concept associations already present in Text-to-Image models rather than enhancing or facilitating the generation of harmful content. By surfacing these associations explicitly, our framework supports increased transparency, interpretability, and responsible auditing. Rather than enabling exploitation, the insights gained help users identify and mitigate biases proactively. Therefore, we view our approach primarily as a protective mechanism—empowering users to better understand and address unintended biases rather than introducing new avenues for misuse.

---

### Review · Reviewer_rxwX · 2025-02-22

**Summary Of Contributions:**

The paper introduces a flexible, concept-driven framework called Concept2Concept. The framework aims to audit text-to-image (T2I) models at scale. By sampling a distribution of prompts and detecting concepts in the generated images, the framework provides a tool (or, measurement) that humans can interpret the behavior of black-box T2I models.
Key contributions include:
* A unified T2I investigation framework including metrics.
* Empirical case studies and findings on the widely used datasets.
* Open-source interactive tool that can be easily used for a broader spectrum of users.

**Audience:**

Yes

**Broader Impact Concerns:**

The paper addressed ethical concerns and limitations in the Appendix. I think the paper tries to carefully handle the potential risks of the paper and its applications.

**Claims And Evidence:**

No

**Requested Changes:**

* Please see the weaknesses above. In particular, please address the prompt/concept overlap more explicitly, and expand the discussion of the ‘concept’ scope. Please connect the proposed measurements and support the importance of 'distributions' with empirical analyses.
* For equation (3), I think P(C) should be represented as a double integral of (t) and (x) as below. Please check and modify if necessary. $p( C ) = \int_t \int_x p(C|x)p_G(x|t)p(t)dxdt$
* Inconsistent reference styles – especially for page 11.

**Strengths And Weaknesses:**

[Strengths]
* **Flexible, Concept-Centered Framework**: By focusing on interpretable concepts and their distributions, the proposed framework offers a versatile tool to investigate T2I models. Each component can be picked out of the box and refined for different objectives.
* **Scalable Approach**: Through automated analysis of large collections of generated images, the framework has strong potential to be used for industrial-scale and production settings. As the authors mentioned, given the widespread deployment of T2I models, the paper is well-timed.
* **Empirical Findings and Insights Through Widely Used Datasets**: The case studies on the Pick-a-Pic and synthetic ImageNet datasets highlight practical usability. The paper identified problematic/undesired results that might easily remain hidden under causal inspection.

[Weaknesses]
* **Reliance on Object Detectors**: The proposed framework critically depends on external object detectors to identify and localize concepts. These “learnable” detectors themselves may exhibit biases learned during training. Also, these detectors, either with “closed” or “open” vocabulary support, would only localize the concepts/terms they learned.
* **Manually Designed Prompt Distributions**: The framework relies on user-defined or manually curated prompts, which could unintentionally input incomplete or biased audits. This also belongs to the problem that the paper is trying to solve – A more robust approach might offer a more representative view of the model’s behavior in practical settings.
* **Limited Breadth of Concept**: While the paper defines “concept” to include object names, nouns, labels, etc., these are somewhat primitive and cannot represent more abstract and semantically meaningful information. For example, there would be difficulties in capturing stylistic attributes, cultural references, overall tone, or mood. I acknowledge that the authors defined the “concept” as visually identifiable features, and they are very important; however, it would be beneficial to discuss this issue.
* **Disconnection of Proposed Metrics**: Although the authors introduce “Concept Frequency”, “Concept Stability”, and “Concept Co-Occurrence”, these terms are barely used in the later context and analyses. In the case studies, the presented figures and discussions employ more generic descriptors (“probability”, “proportion of detections”, “mean co-occurrence count”) instead of explicitly referring to the introduced metrics.
* **Text-Image Concept and Clarification**: The term “Concept2Concept (C2C)” implies an emphasis on relationships among concepts within generated images. However, much of the analysis deals with prompt-to-concepts. In Section 4.1, the paper connects the “prompt” (jogging, running, sprinting) to “concept” (woman, man, glasses, etc.). Here, “jogging” is included in the prompt and not detected by the visual detector, thus confusing to be interpreted as concepts defined in this paper. In this perspective, the sentence “concepts are resolved as we desire” can be somewhat misleading. This creates a potential gap between ‘prompt concepts’ and ‘image concepts.’ Clarifying this would help mitigate confusion. I generally understand the intention of the framework, but it should be clear; for example, one can ask whether those concepts in the prompts (which are not clearly described in the manuscript) and concepts in the generated images are assumed to be in the same set.
* **Necessity of the Full Framework for Dataset Auditing**: In section 5.1, C2C is utilized to uncover problematic or harmful content in the Pick-a-Pic dataset. However, it seems that merely applying an object detector to the images themselves could arguably detect the same content. A more detailed explanation/analysis of why integrating the proposed C2C is necessary for the purpose would strengthen the rationale for adopting this framework.
* **Unclear Value of Concept Distributions**: While the paper emphasizes the importance of ‘distribution-level’ analyses, it remains somewhat unclear how these distributions directly enhance the interpretability or add values to analyses. Please consider establishing a clearer link between the concept “distributions” to various tasks discussed in the paper.

---

> ### Author Response · Authors · 2025-03-07
> **Individual Comments**
>
> Thank you for your feedback. We have addressed your other points in the shared responses above. Below we address concerns that were not shared by other reviewers:
>
> **Text-Image Concept and Clarification**
>
> We appreciate this insightful point. The term "Concept2Concept (C2C)" might suggest an unintended emphasis on solely visual concept relationships. We are open to renaming the framework to avoid confusion. Our original intent behind the naming was to broadly capture the investigation of concept associations in Text-to-Image (T2I) models from all directions—encompassing relationships among concepts present in prompts, detected visually within images, and across prompts and generated outputs. We acknowledge that explicitly differentiating between "prompt concepts" and "image concepts" could enhance clarity, as concepts mentioned in the prompt are not always visually detected explicitly by the detector (as noted in your example).
>
> **Necessity of the Full Framework for Dataset Auditing**
> Applying an object detector alone might appear sufficient to detect problematic or harmful content. However, our framework provides critical additional context through its explicit emphasis on the proposed metrics, including concept co-occurrence. Detecting individual concepts, such as 'naked,' in isolation may not always be indicative of an issue. However, C2C specifically examines how concepts co-occur,  revealing combinations and contexts that alone may be overlooked. For instance, the repeated co-occurrence of 'girl' with 'naked' or ‘dreadlocks’ with ‘beanie’, ‘mask’, and ‘ski’ concepts is crucial to recognizing potentially harmful or misaligned content. Thus, the integration of the framework facilitates a deeper understanding of the contextual nuances within detection results.
>
> **Equation 3**
> You are correct, we displayed the unexpanded form (i.e. plugging in P(C|t) into equation 3.

---

### Review · Reviewer_BdUc · 2025-02-24

**Summary Of Contributions:**

The paper introduces **Concept2Concept**, an auditing framework designed to provide interpretable characterizations of the conditional distribution of images generated by text-to-image models. The core idea is to extract high-level, human-understandable concepts from generated images using **open-vocabulary object detectors** and then analyze these via quantitative metrics such as **concept frequency, stability, and co-occurrence**. By formalizing metrics like concept frequency, stability, and co-occurrence, the authors provide a systematic method to reveal unexpected biases, misalignments, and even harmful content (e.g., CSAM, hyper-sexualization) in both model outputs and training/prompt datasets.  Furthermore, the paper contributes an open-source, interactive visualization tool is provided to facilitate human-in-the-loop auditing, making the framework accessible to non-technical end users.

The authors demonstrate the utility of their framework across several case studies:
- Using a pedagogically designed prompt set to expose associations such as gender biases (e.g., “jogging” paired predominantly with one gender) and exploring nuanced differences in representations (e.g., attire differences between jogging and running).
- By replicating experiments from StableBias and TBYB, the framework shows its ability to provide quantitative insights into gender representation in generated images.
- The framework surfaces stereotypical associations (e.g., over-reliance on wheelchair imagery for disability prompts) and demonstrates how prompt revision can adjust conceptual outputs.
- Applying the method to prompt datasets (such as Pick-a-Pic) and synthetic datasets (StableImageNet) reveals unexpected and harmful content—including the disturbing presence of CSAM and hyper-sexualization—which has critical safety implications.

**Audience:**

Yes

**Broader Impact Concerns:**

The paper addresses a crucial aspect of AI ethics by exposing biases and harmful content in T2I models, which has significant societal implications. However, the detection of sensitive material such as CSAM and hyper-sexualized content—especially involving minors—necessitates a cautious approach. The authors acknowledge these ethical concerns, yet further discussion is needed on:
- [Q1]  How can the framework and its interactive tool be safeguarded against potential misuse or misinterpretation of its findings?
- [Q2]  What protocols should be in place to act upon the detection of harmful content, and how can these audits be integrated into existing safety pipelines without causing undue alarm?

**Claims And Evidence:**

Yes

**Requested Changes:**

**Major revisions**:
- [R1] Expand the discussion on the potential biases and limitations introduced by the object detectors (Florence 2, BLIP VQA) and consider suggesting complementary methods or calibration techniques.
- [R2] Provide additional experiments or ablation studies to analyze the sensitivity of the framework with respect to key hyperparameters and the choice of object detector.
- [R3] Incorporate experiments using more diverse, real-world prompt distributions to demonstrate the framework’s robustness and generalizability.
- [R4] Elaborate on how the insights gained from auditing (e.g., identification of CSAM or misaligned synthetic data) can inform concrete mitigation strategies or adjustments in model training pipelines. This point is critical for strengthening the work’s practical impact.

**Minor revisions**:
- [MR1] Provide a more intuitive explanation of the key metrics (frequency, stability, co-occurrence) along with illustrative examples that guide the reader through their interpretation.
- [MR2] Consider adding a user study or qualitative feedback from non-technical end users who have interacted with the visualization tool to further validate its usability.

**Strengths And Weaknesses:**

**Strengths**:
- [S1] The paper presents a novel, interpretable auditing framework that rigorously defines and analyzes the conditional distribution of generated images in terms of high-level concepts.
- [S2] The authors situate their work well within the context of existing literature, showing how Concept2Concept generalizes and extends prior bias auditing methods.
- [S3] Multiple detailed experiments (e.g., reproducing gender bias findings, auditing disability representation, and detecting harmful content in the Pick-a-Pic dataset) effectively demonstrate the framework’s versatility and practical utility.
- [S4] The provision of an interactive, open-source visualization tool lowers the technical barrier for auditing and encourages broader adoption and collaboration.

**Weaknesses**:
- [W1] The framework’s performance is closely tied to the reliability of off-the-shelf object detectors, which themselves may introduce biases or limitations.
- [W2] As the number of detected concepts grows, especially with high-resolution images or large prompt spaces, the co-occurrence analysis may become computationally intensive.
- [W3] Some experimental setups rely on carefully crafted prompt distributions that might not generalize across different use cases or more naturalistic settings.
- [W4] Some sections, particularly those detailing hyperparameter choices and experimental setups, are dense and could benefit from clearer exposition to improve reproducibility. Certain methodological details, particularly regarding the quantitative metrics and their interpretations, could benefit from clearer explanations and additional visual aids.
- [W5] While the paper effectively identifies problematic biases and associations, it offers only a preliminary discussion on how these findings might translate into actionable mitigation strategies.

---

> ### Author Response · Authors · 2025-03-07
> **Individual Comments**
>
> Thank you for your feedback. We have addressed your other points in the shared responses above. Below we address your comments that were not shared by other reviewers:
>
> **[R4] Elaborate on how the insights gained from auditing (e.g., identification of CSAM or misaligned synthetic data) can inform concrete mitigation strategies or adjustments in model training pipelines. This point is critical for strengthening the work’s practical impact.**
>
> Response: The insights derived from our auditing framework can directly inform practical and actionable mitigation strategies. In the manuscript, we illustrate this with an explicit example: using detected concepts to perform targeted prompt revision. Since our framework provides interpretable characterization of detected concepts, practitioners can selectively attenuate or amplify certain concepts through positive or negative prompting. This method is detailed in Experiment 3, specifically shown in Figure 4 of the manuscript.
>
> Additional concrete mitigation strategies enabled by our auditing insights include:
> - Data pruning: Utilizing the identification of problematic concepts (such as sensitive or undesirable content like CSAM) to exclude or prune datasets prior to model training.
> - Contextualization of prompts (labels): When synthetic or misaligned data are identified, additional context or clarification can be appended to class labels to ensure alignment between the input concepts and intended user interpretations.
>
> These approaches highlight the direct practical benefits of our framework, showcasing its capability to drive meaningful adjustments in training pipelines and ultimately enhance model reliability and alignment.

---

### Author Response · Authors · 2025-02-27
**Thank you, We Are Addressing Your Feedback**

Dear Reviewers,
Thank you for your valuable feedback. We appreciate your insights and are actively working to address your main shared concerns as detailed below. For concerns that were not shared by 2 or more reviewers, we will address them individually. In the meantime, please let us know if this aligns with your expectations and whether our plan adequately addresses your concerns.

**Change: Ablation study on the object detector**

**Reviewers’ Request:**

- [BdUc] [R2] Provide additional experiments or ablation studies to analyze the sensitivity of the framework with respect to key hyperparameters and the choice of object detector. [R1] Expand the discussion on the potential biases and limitations introduced by the object detectors (Florence 2, BLIP VQA) and consider suggesting complementary methods or calibration techniques.

- [rxwX] Reliance on Object Detectors: The proposed framework critically depends on external object detectors to identify and localize concepts. These “learnable” detectors themselves may exhibit biases learned during training. Also, these detectors, either with “closed” or “open” vocabulary support, would only localize the concepts/terms they learned.

- [fziz] The analysis relies heavily on the object detection models. Although the authors have mentioned in the Limitations section, they do not specify clearly how to mitigate the influence. The authors could report human evaluation on (a small portion of) images analyzed in this study. Additionally, the choice of detectors needs to be well-justified, and, adding comparison with alternative detectors will be good. The detectors may not work well on non-photorealistic images, such as cartoon, abstract, surreal, or artistic images.

**Change:  Address cultural influence**

**Reviewers’ Request:**

- [rxwX]   Limited Breadth of Concept: While the paper defines “concept” to include object names, nouns, labels, etc., these are somewhat primitive and cannot represent more abstract and semantically meaningful information. For example, there would be difficulties in capturing stylistic attributes, cultural references, overall tone, or mood. I acknowledge that the authors defined the “concept” as visually identifiable features, and they are very important; however, it would be beneficial to discuss this issue.

- [fziz] Concept associations are analyzed without considering the broader context of the image or prompt. For example, detecting a "thong" or "naked" might be appropriate in some artistic or cultural contexts but is flagged as problematic.

**Change:  Prompt Distribution and Generalization**

**Reviewers’ Request:**

- [BdUc] [W3] Some experimental setups rely on carefully crafted prompt distributions that might not generalize across different use cases or more naturalistic settings. [R3] Incorporate experiments using more diverse, real-world prompt distributions to demonstrate the framework’s robustness and generalizability.

- [rxwX]  Manually Designed Prompt Distributions: The framework relies on user-defined or manually curated prompts, which could unintentionally input incomplete or biased audits. This also belongs to the problem that the paper is trying to solve – A more robust approach might offer a more representative view of the model’s behavior in practical settings.

- [fziz] Some experiments lack generalizability. For example, section 4.1 studies only three actions on one model (SD2.1). Readers are not sure whether the conclusion can be generalized to a broader range of concepts and models.

**Change:  Clearer exposition on proposed metrics**

**Reviewers’ Request:**

- [BdUc] [MR1] Provide a more intuitive explanation of the key metrics (frequency, stability, co-occurrence) along with illustrative examples that guide the reader through their interpretation.

- [rxwX] Disconnection of Proposed Metrics: Although the authors introduce “Concept Frequency”, “Concept Stability”, and “Concept Co-Occurrence”, these terms are barely used in the later context and analyses. In the case studies, the presented figures and discussions employ more generic descriptors (“probability”, “proportion of detections”, “mean co-occurrence count”) instead of explicitly referring to the introduced metrics.

---

> ### Comment · Reviewer_BdUc · 2025-03-01
>
> I appreciate the authors for summarizing the shared concerns among reviewers and for the clear plan for revision. Looks good to me. Looking forward to the revised version.

---

### Author Response · Authors · 2025-03-07
**Reviewers: [BdUc][rxwX][fziz] Requested Change: Ablation Study on Object Detector**

We greatly appreciate the reviewers' feedback regarding the importance of clearly articulating the sensitivity and limitations of our choice of object detector. **Following the reviewers' suggestions, we have expanded our analysis by conducting an ablation study and further clarifying our selection rationale. The full experimental details can be found in the manuscript (Section A.5 Figures 8 and 9).** We summarize it below:

We performed an ablation study using the prompt set from the experiment 1 (~3k images)  involving five detectors: Florence 2, Kosmos, GroundingDINO, OwlV2, and DETR. We included DETR, despite recognizing its limitations as a closed-vocabulary detector restricted to classes from its training set, explicitly to demonstrate the rationale behind excluding such closed-vocabulary models from our framework.

Our results indicate the following important findings: first, Florence-2 provides the best performance (most unique detected concepts) and smallest model size (i.e. faster and more diverse inference). Second, closed vocabulary models like DETR--which was trained on the class labels of the COCO dataset-- are not useful for broad concept detection. Third, although both GroundingDino and OwlV2 rely on the same BLIP caption, they do exhibit clear differences in their grounding performance (as indicated by Figures 8 and 9 in section A5.}
Our selection of Florence-2 was informed by its strong balance between size, speed, and detection capabilities.  Florence 2 offers the advantage of being smaller and faster than Kosmos-2. Additionally, Florence-2 qualitatively outperforms other notable models such as CLIP, SAM, Flamingo, and even Kosmos-2, as documented in the Florence 2 paper. Models like CLIP and Flamingo are unsuitable for our framework due to their inability to localize concepts, and SAM, while effective at localization, lacks semantic labeling.

We agree with the reviewers that the reliance on object detectors inherently introduces biases and limitations. Given the absence of ground truth annotations for detected concepts, we explicitly designed Concept2Concept (C2C) to support human-in-the-loop auditing, allowing users to verify and contextualize detections through bounding boxes which enable visual inspection and concept co-occurrence information. This design ensures that even when detectors produce false positives or miss certain concepts, users are empowered to critically assess and interpret the results.

We also want to note that specialized detectors (such as OCR for numbers or NSFW content detectors) will inherently differ significantly in their capabilities and scope than generalist models. Our framework is intentionally designed with detector interchangeability in mind, requiring only that detectors provide (1) open-set detection capabilities and (2) localization functionality. Indeed, specialized versions of Florence 2 have already been released by its authors for domain-specific tasks.

We sincerely thank the reviewers for their valuable suggestions, which have strengthened our manuscript and clarified both our methodology and its practical considerations.

---

### Author Response · Authors · 2025-03-07
**Reviewers: [rxwX][fziz] Requested Change: Address Concept Scope**

To reviewer [rxwX]: addressing the scope of concept representation.

*First, we have integrated a summarized version of your feedback to the paper. Please note that revisions are in pink/magenta color in the revised manuscript. Thank you for your feedback!*

We acknowledge that stylistic attributes would indeed be relevant for certain applications, such as copyright detection where a prompt concept maps onto a specific artistic style. However, grounding such attributes within an image remains a technical challenge, as stylistic elements often do not correspond to discrete, localized entities but rather emerge holistically from multiple compositional factors.

That being said, our framework integrates Florence 2 which actually does capture some mood-related concepts through descriptive captions (which are then grounded). For example:

“The image shows a young woman jogging on a paved path in a park. She is wearing a gray tank top and black leggings and white sneakers. The path is lined with trees on both sides, with their leaves changing colors, indicating that it is autumn. The sky is blue and the sun is shining through the trees, casting a warm glow on the scene. The woman appears to be in motion, with her arms and legs pumping as she runs. The grass on the left side of the path is green and well-maintained, and there are a few buildings visible in the background. The overall mood of the image is peaceful and serene.”

However, these descriptors are difficult to precisely localize within an image (e.g., how do we assign “peaceful” to a specific region of the image?) and thus do not align well with our framework, which relies on concept grounding and interpretability.

Instead of treating mood or style as indivisible wholes, we argue that these abstract qualities can already be inferred through the existing detected concepts in our framework. For instance, the mood of "peaceful and serene" may emerge from a combination of autumn trees, warm sunlight, green grass, and open paths—all of which are identifiable and detectable within our current approach. We enable users to analyze the underlying visual elements that contribute to them, making these abstract qualities interpretable through concept associations. We leave it to future work to explore how a similar decomposition approach could be applied to artistic styles, such as those of Van Gogh—specifically, what kinds of visual concepts (e.g., color palettes, composition structures) would need to be detected in order to infer a specific artistic style.

Furthermore, we emphasize that our definition of "concept" follows established prior work in bias auditing and interpretability (as detailed in Section 1 and 2). Our framework does not redefine the concept space but rather builds on previous methodologies that have effectively used object-based and semantic category-based characterizations.

---

> ### Author Response · Authors · 2025-03-07
> **(continued)**
>
> To reviewer [fziz]: addressing contextual considerations in concept associations.
>
> First, we have integrated a summarized version of your feedback to the paper. Please note that revisions are in pink/magenta color in the revised manuscript. Thank you for your feedback!
>
> In response to your concern that our analysis does not sufficiently consider the broader context of the image or prompt, we argue that context is a core component of our framework in multiple ways:
>
> 1. Localization of Concepts within Images: our framework does not simply report concept presence alone—we localize each concept within the generated image, allowing for direct inspection of how and where it appears. This provides rich visual context that helps users interpret potential biases or misalignments.
> 2. Concept-to-Concept Relationships: the very name Concept2Concept reflects our intention to capture not just concept occurrences, but their relationships—specifically, how concepts in the prompt (a key source of context) lead to detected concepts in the generated images. This structure inherently provides broader semantic grounding, connecting input text concepts to visual outputs rather than treating detected concepts in isolation.
> 3. Concept Co-Occurrence: to explicitly capture contextual relationships, we introduced concept co-occurrence metrics. This allows us to investigate which concepts systematically appear together, helping to reveal patterns that might indicate bias or unintended associations.
>
> Regarding the nudity and thong examples, we wish to clarify our intention: we do not argue that nudity or revealing clothing is always inherently harmful or inappropriate. What is problematic is when these associations occur disproportionately for specific marginalized groups—for instance, if women and girls are overrepresented in such imagery, this reinforces harmful hypersexualized biases in AI-generated outputs.  The significance of this issue is heightened in datasets that are not merely research curiosities but are actively used in downstream AI applications, such as training evaluation frameworks (e.g., PickScore) or serving as publicly available resources for future model development (e.g., Pick-a-Pic and StableImageNet). This concern is further amplified by the common assumption among researchers that such datasets have already undergone moderation before being released, which may not always be the case. If these biases are not audited and addressed, they risk perpetuating harmful stereotypes at scale.
>
> Furthermore, the importance of addressing nudity and related content in AI-generated imagery is widely recognized in the field, as evidenced by the prevalence of NSFW filtering in T2I research and industry applications [1-6]. Our framework is not about making moral judgments but rather providing a systematic approach to understanding and characterizing these associations, ensuring that any (un)intended patterns can be identified and addressed in a transparent manner.
>
> Sources:
>
> [1] Li, Xinfeng, et al. "Safegen: Mitigating sexually explicit content generation in text-to-image models." Proceedings of the 2024 on ACM SIGSAC Conference on Computer and Communications Security. 2024.
>
> [2] Poppi, Samuele, et al. "Safe-CLIP: Removing NSFW concepts from vision-and-language models." European Conference on Computer Vision. Cham: Springer Nature Switzerland, 2024.
>
> [3]  Microsoft Azure https://learn.microsoft.com/en-us/azure/ai-services/openai/concepts/content-filter?tabs=warning%2Cuser-prompt%2Cpython-new
>
> [4] Amazon AWS Comprehend  https://docs.aws.amazon.com/comprehend/latest/dg/trust-safety.html
>
> [5] OpenAI-Moderation  https://platform.openai.com/docs/guides/moderation
>
> [6] OpenAI-Moderation Markov, Todor, et al. "A holistic approach to undesired content detection in the real world." Proceedings of the AAAI Conference on Artificial Intelligence. Vol. 37. No. 12. 2023.

---

### Author Response · Authors · 2025-03-07
**Reviewers: [BdUc][rxwX][fziz] Requested Change: Clarify Prompt Distribution and Generalization**

We appreciate the reviewers’ concerns regarding the generalization of our prompt distributions and whether manually curated prompts might limit the applicability of our findings. We would like to clarify that our study employs a diverse set of prompt distributions that go beyond hand-crafted examples. While we do use a small set of pedagogically designed prompts in one case study, the rest of our experiments leverage real-world datasets from prior works, ensuring broad applicability and alignment with existing research.

The *only* manually designed prompts in our work are in Case Study 1: Pedagogically Designed Prompt Set. These prompts are intentionally controlled to isolate specific biases and *serve as an illustrative example* of how Concept2Concept can surface biases. This does not reflect a limitation of the framework but rather an intended demonstration of its capabilities in a controlled setting.

The remaining case studies rely on real-world datasets:
- Case Study 2 (Reproducing Bias Probing Results) uses prompts from StableBias and TBYB, which are widely used in prior literature for bias audits.
- Case Study 3 (Scaling Up Disability Representation Analysis) expands on qualitative findings from previous work to systematically quantify biases using a diverse and empirically driven prompt set.
- Case Study 4 (Auditing Pick-A-Pic Dataset) investigates a real-world dataset of user-generated prompts, which are directly reflective of how people interact with T2I models in practice. The original dataset has 0.5M instances and 35K distinct prompts.
- Case Study 5 (Auditing StableImageNet) examines a synthetic dataset generated using class labels as prompts, mimicking structured dataset curation practices in AI model training. StableImageNet has 1k classes since it is based on the ImageNet1K dataset.

**To integrate your feedback, we have included a clearer exposition in the methods section 3 and appendix section B (table 2), clarifying how Concept2Concept generalizes to various user cases.** We have structured our experiments to demonstrate a range of use cases for Concept2Concept.  Rather than applying the framework to a single dataset or use case, we show how it can be  adapted to different auditing goals.  To make it clearer how Concept2Concept can be used across different auditing needs, we also provide a structured guide outlining how users can apply our framework based on their specific goals in Appendix Table 2. Thank you for your feedback!

---

### Author Response · Authors · 2025-03-07
**Reviewers: [BdUc][rxwX] Requested Change: Clearer Exposition on Proposed Metrics**

To Reviewer [BdUc]:
We appreciate the suggestion to provide a more intuitive explanation of our key metrics—Concept Frequency, Concept Stability, and Concept Co-Occurrence—and to include illustrative examples that aid interpretation. Initially, due to space constraints, we had moved our figure-based method overview (which contained these examples) to the appendix. However, we agree that having it in the main text improves readability and accessibility. **We have now relocated this figure to the main text and revised the surrounding discussion to better guide the reader through the interpretation of each metric.**  To accommodate this change while adhering to the 12-page limit, we have moved a portion of the Related Work section to the appendix.


To Reviewer [rxwX]:
Thank you for pointing out the disconnect between the introduced metrics and their later usage in the figures and discussions. We acknowledge that while we formally introduced Concept Frequency, Concept Stability, and Concept Co-Occurrence, the case studies at times used more generic terms such as “probability,” “proportion of detections,” and “mean co-occurrence count,” which may have led to confusion.

To be consistent, we have:

- **Updated our figures and captions to explicitly reference the introduced metrics. We keep “mean co-occurrence count” in Figure 5 as this is an average over 10 samples each of size N. Because N is large, we report the count or unnormalized P(c,c’) (i.e. P(c,c’)xN)**
- **Revised the text in case study discussions to maintain clear alignment with the defined terms.**

---

### Author Response · Authors · 2025-03-07
**Response Structure**

Dear Reviewers,

Thank you for your detailed responses. We have carefully incorporated your feedback. In the revised manuscript, all revisions are highlighted in pink/magenta for easy reference. We first address the shared concerns using a "respond all" approach through the official comment. For concerns that are not shared, we provide individual responses to each reviewer. Your insights have been invaluable in enhancing the clarity and rigor of our work.

Thank you again for your time and thoughtful feedback!

The Authors

---

### Decision · Action_Editor_7gSC · 2025-04-06

**Recommendation:** Accept as is

**Comment:**

The authors answered questions and addressed the concerns raised by the reviewers through updates in the manuscript.
Though the presented case-studies are seen to be more of an illustrative nature rather than a comprehensive study conclusively validating the generalizability of the framework, and despite some remaining uncertainties related to the dependence on a concept detector (with potentially its own biases), a prompt distribution (may be difficult to construct) and discussion of cultural nuances in sensitivity to concepts, the proposed framework is introduced in an accessible manner and has the potential to trigger the attention of the relevant part of the community.

**Audience:**

The paper introduces a rather simple practical concept-based framework for auditing text-to-image models in terms of their biases or stereotypes. The framework can assist researchers developing such models or using such models for downstream tasks. Using the framework the paper also points out potential issues in existing datasets and models which should be of interest to users of these.

**Claims And Evidence:**

The paper demonstrates the applicability of the proposed framework on a series of case studies. It acknowledges its dependence on an external concept detector and discusses some choices and their effects within an ablation study. It also provides some guidance on setting up a task-relevant prompt distributions, which admittedly is one of the critical bottlenecks of the framework.